# Two KaiABC systems control circadian oscillations in one cyanobacterium

Christin Köbler [1,5], Nicolas M. Schmelling [2,5], Anika Wiegard [2,5], Alice Pawlowski [2,5], Gopal K. Pattanayak[3], Philipp Spät [4], Nina M. Scheurer[1], Kim N. Sebastian[1], Florian P. Stirba[2], Lutz C. Berwanger[2], Petra Kolkhof[2], Boris Maček [4], Michael J. Rust [3], Ilka M. Axmann [2] ✉ & Annegret Wilde [1] ✉

The circadian clock of cyanobacteria, which predicts daily environmental changes, typically includes a standard oscillator consisting of proteins KaiA, KaiB, and KaiC. However, several cyanobacteria have diverse Kai protein homologs of unclear function. In particular, *Synechocystis* sp. PCC 6803 harbours, in addition to a canonical *kaiABC* gene cluster (named *kaiAB1C1*), two further *kaiB* and *kaiC* homologs (*kaiB2*, *kaiB3*, *kaiC2*, *kaiC3*). Here, we identify a chimeric KaiA homolog, named KaiA3, encoded by a gene located upstream of *kaiB3*. At the N-terminus, KaiA3 is similar to response-regulator receiver domains, whereas its C-terminal domain resembles that of KaiA. Homology analysis shows that a KaiA3-KaiB3-KaiC3 system exists in several cyanobacteria and other bacteria. Using the *Synechocystis* sp. PCC 6803 homologs, we observe circadian oscillations in KaiC3 phosphorylation in vitro in the presence of KaiA3 and KaiB3. Mutations of *kaiA3* affect KaiC3 phosphorylation, leading to growth defects under both mixotrophic and chemoheterotrophic conditions. KaiC1 and KaiC3 exhibit phase-locked free-running phosphorylation rhythms. Deletion of either system (Δ*kaiAB1C1* or Δ*kaiA3B3C3*) alters the period of the cellular backscattering rhythm. Furthermore, both oscillators are required to maintain high-amplitude, self-sustained backscatter oscillations with a period of approximately 24 h, indicating their interconnected nature.

Three genes, *kaiA*, *kaiB*, and *kaiC*, encode the core circadian oscillator in Cyanobacteria[1]. Over the last few decades, the biochemical interplay between these three proteins has been studied in great detail in *Synechococcus elongatus* PCC 7942 (hereafter *Synechococcus*). The KaiC protein forms a homohexamer and has autokinase, autophosphatase, and ATPase activities[1–4]. By associating with KaiC, KaiA stimulates the autokinase and ATPase activities of KaiC, and thus, the protein gets phosphorylated[5–7]. Upon phosphorylation of two neighboring residues (Ser431 and Thr432), KaiC undergoes structural rearrangements,

exposing a binding site for KaiB[8–10]. After binding, KaiB sequesters KaiA from KaiC, promoting KaiC's autophosphatase activity, and the protein reverts back to its unphosphorylated state[8,9,11]. The interplay between KaiA and KaiB is crucial for the KaiC phosphorylation cycle, which confers clock phase and rhythmicity to the cell[12,13]. For a more detailed review on the KaiABC oscillator and its regulatory network, see Cohen and Golden[14], Swan et al.[15] and Snijder and Axmann[16].

Although most studies on prokaryotic circadian rhythms have focused on the cyanobacterium *Synechococcus*, the standard KaiABC

[1]Institute of Biology III, Faculty of Biology, University of Freiburg, 79104 Freiburg, Germany. [2]Institute for Synthetic Microbiology, Biology Department, Heinrich Heine University Düsseldorf, 40225 Düsseldorf, Germany. [3]Department of Molecular Genetics and Cell Biology, The University of Chicago, Chicago, IL 60637, USA. [4]Department of Quantitative Proteomics, Interfaculty Institute for Cell Biology, Eberhard Karls University Tübingen, 72076 Tübingen, Germany. [5]These authors contributed equally: Christin Köbler, Nicolas M. Schmelling, Anika Wiegard, Alice Pawlowski. ✉e-mail: Ilka.Axmann@hhu.de; annegret.wilde@biologie.uni-freiburg.de

system is functionally conserved in other cyanobacteria[17]. In addition to the standard KaiABC system, divergent homologs of KaiB and KaiC have been identified in cyanobacteria, other bacterial species, and archaea[18]. The structure, mechanism of function, and physiological roles of these homologs are often unclear. A few studies have demonstrated the role of KaiB and KaiC homologs in stress responses in e.g. *Legionella pneumophila*[19] and *Pseudomonas* species[20]. Other Kai homologs are also involved in the regulation of diurnal rhythms outside the cyanobacterial lineage. These include e.g. KaiB and KaiC homologs from the phototrophic bacterium *Rhodopseudomonas palustris*[21]. Recently, a KaiA-independent hourglass timer was reconstituted using *Rhodobacter sphaeroides* (*Rhodobacter*) KaiC and KaiB homologs[22]. *Rhodobacter* KaiC exhibits a divergent extended C-terminus that is typically found in proteins belonging to the KaiC2 subgroup. This C-terminal extension mediates hexamer-hexamer interactions, allowing KaiA-independent phosphorylation. *Rhodobacter* KaiB controls the phosphorylation-dephosphorylation cycle of KaiC depending on the ATP-to-ADP ratio, suggesting that metabolic changes during the day and night cycles drive this KaiBC clock[22] as it has been previously shown for *Synechococcus* KaiABC[23].

The cyanobacterium *Synechocystis* sp. PCC 6803 (*Synechocystis*) is a facultative heterotrophic cyanobacterium that, unlike *Synechococcus*, can utilize glucose as an energy and carbon source. In addition to the canonical *kaiAB1C1* gene cluster, *Synechocystis* encodes two further *kaiB* homologs, *kaiB2* and *kaiB3*, and two *kaiC* homologs, *kaiC2* and *kaiC3*[24]. For the *Synechocystis* KaiB3-KaiC3 timing system, Aoki and Onai suggested a function in the fine-tuning of the core oscillator KaiAB1C1 by modulating its amplitude and period[25]. This idea was supported by Wiegard et al., who investigated the characteristics of the KaiC3 protein and proposed an interplay between the KaiB3-KaiC3 system and the proteins of the standard clock system[26]. Furthermore, autophosphorylation and ATPase activities of *Synechocystis* KaiC3 have been verified, suggesting that enzymatic activities might be conserved across the KaiC protein family[26–28]. Recently, Zhao et al.[17] used a luminescence gene reporter to study circadian gene expression in the *Synechocystis* wild type in comparison to mutant strains lacking each of the *kai* genes. They demonstrated that *kaiAB1C1* and *kaiB3C3* genes are both important for circadian rhythms in *Synechocystis*, whereas *kaiC2* and *kaiB2* deletion mutants still showed rhythmic gene expression, which is in agreement with previous suggestions by Aoki and Onai[25]. Phenotypic mutant analysis by our group revealed that two systems function in the autotrophy/heterotrophic switch, especially affecting heterotrophic growth. In contrast to the study by Zhao et al.[17] that reported a small growth defect in *Synechocystis* in which *kaiC3* has been deleted, our studies with the motile *Synechocystis* strain (PCC-M[29]) showed no such effect on growth under light/dark (LD) cycles. However, the mutant strain displayed a growth defect under chemoheterotrophic conditions in the dark compared with the wild type[26,30]. This impairment was less severe in comparison with the Δ*kaiAB1C1*-deficient strain, which completely lost its ability to grow in the dark. Notably, the complete deletion of *kaiC2* was not possible in the wild-type strain used in our laboratory. Although Zhao et al.[17] clearly showed that deletion of the *kaiC3* and *kaiB3* genes affects the circadian rhythm of *Synechocystis*, it remains unclear whether the KaiB3-KaiC3 system can function as an oscillator. How can such a minimal system maintain circadian rhythmicity without KaiA? *Prochlorococcus* MED4, which lacks a *kaiA* gene in its entire genome, lacks free-running circadian rhythmicity[31,32]. Moreover, *Synechocystis* KaiC3 lacks the extended C-terminus, which is crucial for the oscillation of the *Rhodobacter* KaiBC hourglass timer[22].

In *Synechococcus*, the KaiA protein functions as a homodimer and harbors two distinct domains connected by a linker sequence[33–35]. The N-terminal domain is similar to bacterial response regulators, but lacks the aspartate residue crucial for phosphorylation; hence, it is designated as a pseudoreceiver domain (PsR domain)[33]. This domain was shown to bind the oxidized form of quinones and is therefore able to sense the onset of darkness and forward signals to the C-terminal domain[33,36]. The C-terminus has a four-helix bundle secondary structure and is highly conserved within Cyanobacteria. The domain harbors the KaiA dimer interface and the KaiC binding site, and is necessary to stimulate the autophosphorylation activity of KaiC[33,35]. Mutations in *kaiA* resulting in altered periodicity were mapped throughout both domains, indicating their importance in rhythmicity[35,37].

To date, the regulatory network of the KaiB3-KaiC3 system in *Synechocystis* remains unclear, as it does not interact with KaiA and does not utilize the SasA-RpaA output pathway, suggesting alternative yet unidentified components for KaiB3-KaiC3-based signal transduction[38]. In a large-scale protein-protein interaction screen, a potential interaction partner of KaiC3 was identified[39]. This protein, Sll0485, was categorized as a NarL-type response regulator and could be a potential element in the KaiB3-KaiC3 signaling pathway[40].

In this study, we computationally characterized Sll0485 and detected strong co-occurrences of the KaiB3-KaiC3 system with Sll0485 in the genomic context of Cyanobacteria and other bacteria. Because of this and the fact that its C-terminal domain shares similarities with KaiA homologs, we designated this protein KaiA3. Based on the in vitro analyses, we propose that *Synechocystis* KaiA3 forms an oscillator with KaiB3 and KaiC3. KaiA3 driven phosphorylation rhythms of KaiC3 are phase-locked with KaiC1 phosphorylation in the cell. Both systems, KaiA1B1C1 and KaiA3B3C3, appeared to control circadian rhythmicity and the phototrophy-to-heterotrophy switch in *Synechocystis*.

## Results

### KaiA3 is a chimeric protein harboring a NarL-type response regulator domain at the N-terminus and a conserved KaiA-like motif at the C-terminus

The canonical clock genes, *kaiABC* and *kaiA1B1C1*, form a cluster in *Synechococcus* and *Synechocystis*, respectively. In contrast, the *kaiB3* and *kaiC3* genes of *Synechocystis* are localized in different regions of the chromosome (Fig. S1a). Here, the *kaiB3* gene forms a transcriptional unit with the upstream open reading frame *sll0485 (kaiA3)*. KaiA3 has been annotated as a NarL-type response regulator[40]. Using reciprocal BLAST analyses, we detected orthologs of KaiA3 in 15 cyanobacterial species (16.5% of cyanobacterial species contained at least one KaiC homolog), mainly belonging to the order *Chroococcales*[41] (Supplementary Data S1), and in five bacterial genera outside Cyanobacteria, namely, *Roseiflexus*, *Chloroflexus*, *Chloroherpeton*, *Rhodospirillum*, and *Bradyrhizobium*.

Owing to the genetic context, we aligned the cyanobacterial KaiA3 orthologs with both, a NarL-type response regulator (Fig. S2), and cyanobacterial KaiA proteins (Fig. 1a). The canonical NarL protein consists of an N-terminal receiver domain, a linker, and a C-terminal DNA-binding domain with a helix-turn-helix motif[40,42]. The N-terminus of the KaiA3 orthologs is conserved and indeed shows limited homology to NarL-type response regulators (Fig. S2). However, the similarities to the NarL protein decreased in the C-terminus (Fig. S2). Concurrently, the conservation between KaiA3 and the KaiA protein family increased (Fig. 1a). The conserved residues in the C-terminus correspond to structurally important features of the *Synechococcus* KaiA protein, such as α-helical secondary structures, the KaiA dimer interface, or residues critical for the KaiA-KaiC interaction[33,34] (Fig. 1a). Additionally, the lack of conservation in the N-terminus compared to that observed in known KaiA orthologs is consistent with the results of Dvornyk and Mei, who proposed that different N-terminal domains exist for KaiA homologs for functional diversification[43]. Because of its similarity to KaiA and synteny with the *kaiB3* gene, we named the hypothetical Sll0485 protein KaiA3. Furthermore, to facilitate the

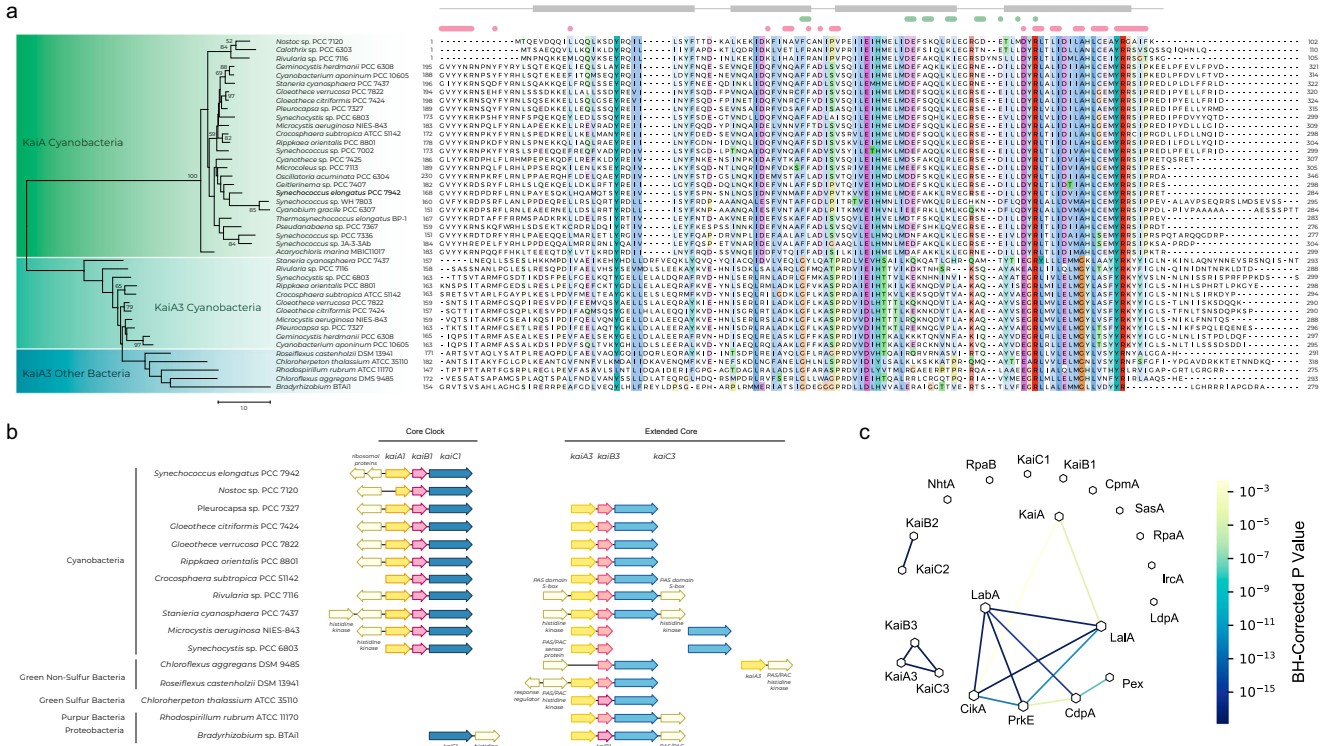

**Fig. 1 | Bioinformatic analyses of Sll0485 (KaiA3). a** Multiple sequence alignment and maximum likelihood-inferred phylogenetic reconstruction of KaiA3 and selected KaiA orthologs. The sequences were aligned with Mafft (L-INS-i default parameters, Jalview), trimmed to position 168 of the C-terminus of *Synechococcus* KaiA and are represented in the Clustalx color code with conservation visibility set to 25%. Marks above the alignment refer to *Synechococcus* KaiA as a reference. Light green bars and dots indicate residues critical for KaiC interaction, light pink bars and dots represent residues important for dimerization, and light gray blocks outline residues forming α-helices as secondary structures. Aligned sequences were used to infer a maximum likelihood protein tree. The scale bar indicates one substitution per position. Bootstrap values (*n* = 1000) are displayed on the branches. Bootstrap values of less than 50 are not shown. **b** Synteny analysis of *kaiA1B1C1* compared to *kaiA3*, *kaiB3*, and *kaiC3* genes for selected bacterial species. Analysis was performed with the online tool SyntTax, a prokaryotic synteny and taxonomy explorer (https://archaea.i2bc.paris-saclay.fr/synttax/; 2020-06-08). Default settings were used for analysis (best match, 10% norm. Blast). **c** Co-occurrence of KaiA3 with circadian clock proteins in cyanobacteria using pairwise right-sided Fisher's exact test. Network of significant co-occurring circadian clock factors from Schmelling et al.[27], including KaiA3 in Cyanobacteria. The line color corresponds to the level of significance resulting from pairwise Fisher's exact test. Missing links were those with p-values higher than 0.01. The node size is proportional to the degree of that node.

distinction of KaiA homologs, we will use the name KaiA1 for the *Synechocystis* KaiA core clock homolog Slr0756.

A gene tree resulting from multiple sequence alignments (Fig. 1a) distinctly separated KaiA3 from the canonical KaiA orthologs. To further investigate the evolutionary relationship of KaiA3, multiple sequence alignments of the C-termini of orthologs of KaiA3, KaiA, and Slr1783 (Rre1) as a reference for NarL orthologs in Cyanobacteria[44] were used to construct a phylogenetic tree (Fig. S3). Here, KaiA3 orthologs form a distinct clade at the basis of the KaiA orthologs when compared to both orthologous groups of Slr1783 (Rre1)/NarL (*E. coli*, UniProtKB - P0AF28) and KaiA simultaneously (Fig. S3). In summary, these findings strengthen the idea that the C-terminus of KaiA3 functions similarly to that of KaiA. We further constructed three-dimensional models of KaiA3 to gain a better understanding of its potential functions. To date, no structure is available for KaiA3, and it is difficult to generate a reliable three-dimensional model covering the full-length KaiA3 sequence because of the enigmatic structure of the linker region, for which no significant similarities could be detected. However, secondary structure prediction suggested that the N-terminus structurally aligns with NarL (Fig. S4a). Therefore, we modeled the N-terminus (residues 1-140) and the remaining part of the sequence separately (residues 141-299). For the N-terminus, numerous hits for response regulator domains were obtained, with *E. coli* NarL (PDB 1A04) showing the highest degree of sequence similarity. The 3D-model structures of KaiA3 are highly similar and display

the canonical fold of response regulator domains: a central five-stranded parallel β-sheet flanked on both faces by five amphipathic α-helices and a phosphorylatable aspartate residue in the β3-strand (Fig. S4b). This aspartate residue (D65) plays a role in response regulator phosphorylation (Fig. S2, blue stars) and is conserved in all species, except *Pleurocapsa* and *Microcystis*. Thus, most KaiA3 homologs, including the *Synechocystis* protein, harbor a potential phosphorylation site. Furthermore, the structure superimposes well on the PsR domain of KaiA, even though the PsR domain lacks the phosphate-accepting aspartate residue and the α4-helix between the β4- and β5-strands (Fig. S4b). The amino acid sequence between the β4- and β5-strands shows the least conservation between KaiA and KaiA3; however, the level of sequence conservation in this region is generally low for KaiA and its homologs[35]. In contrast to the N-terminal response regulator domain, the C-terminal domain of KaiA3 revealed a unique fold, which has only been detected in KaiA thus far[45], and the N-terminal domain of the phosphoserine phosphatase RsbU from *Bacillus subtilis*[46], namely, a unique four-α-helix bundle constituting the KaiA-like motif (Fig. S4c). In conclusion, we propose that KaiA3 consists of two protein modules: (i) the N-terminal domain, resembling a NarL-type response regulator receiver domain, including its phosphorylation site, and (ii) the C-terminal domain displaying features of a KaiA-like motif. This is particularly intriguing because putative *kaiA* orthologs outside Cyanobacteria were not identified until recently[43].

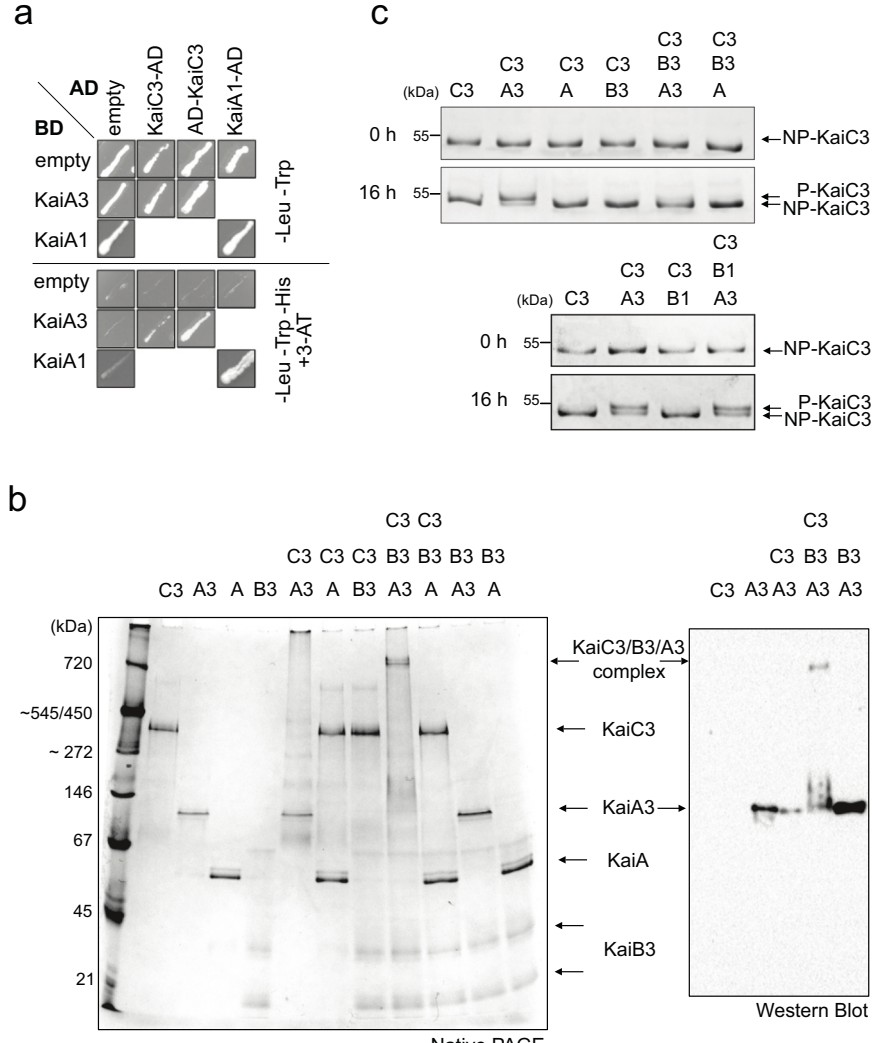

**Fig. 2 | Analysis of KaiA3 protein interactions and KaiC3 phosphorylation.**
**a** YTH interaction analysis between KaiA3 and KaiC3. The KaiA1 dimer interaction was used as a positive control. YTH reporter strains carrying the respective bait and prey plasmids were selected by plating on complete supplement medium (CSM) lacking leucine and tryptophan (-Leu -Trp). AD, GAL4 activation domain; BD, GAL4 DNA-binding domain; empty, bait, and prey plasmids without protein sequence (only AD/BD domain). The physical interaction between bait and prey fusion proteins was determined by growth on complete medium lacking leucine, tryptophan, and histidine (-Leu -Trp -His) and the addition of 12.5 mM 3-amino-1,2,4-triazole (3-AT). The BD was fused to the N-terminus of KaiA3. For clarity, spots were assembled from several replicate assays (the original scans are shown in Fig. S5). KaiC3-KaiA3 interaction analysis was performed thrice. **b** Interaction analysis of recombinant Kai proteins on native polyacrylamide gel. Proteins were incubated for 16 h at 30 °C and subsequently subjected to 4–16% clear native PAGE. Gels were either stained with Coomassie blue (left side) or blotted and immunodecorated with a monoclonal anti-His antibody to detect recombinant KaiA3-His6 (right side). Representative images of three independent experiments. Recombinant *Synechococcus* KaiA was used for comparison. **c** KaiC3 phosphorylation depends on the presence of KaiA3 and KaiB3. KaiC3 was dephosphorylated by incubating for 18 h at 30 °C prior to the start of the assay. 0.2 µg/µl NP-KaiC3 was incubated at 30 °C in the presence or absence of 0.1 µg/µl *Synechocystis* KaiA3 (A3), KaiB3 (B3) and KaiB1 (B1) and *Synechococcus* KaiA (A), respectively. Aliquots were taken at 0 and 16 h, followed by separation on a high-resolution LowC SDS-PAGE gel in Tris-Tricine buffer and staining with Coomassie blue. A slow-migrating band representing the phosphorylated form of KaiC3 (P-KaiC3) was observed only in the presence of KaiA3. NP indicates dephosphorylated KaiC3. Phosphorylation analysis was performed at least three times, and a negative control for *Synechococcus* KaiA was performed twice. Source data are provided as a Source Data file.

## Conserved synteny and co-occurrence of KaiA3 and the KaiB3-KaiC3 system among prokaryotes

As in *Synechocystis*, we found the *kaiA3* gene upstream of *kaiB3* in all the analyzed cyanobacterial genomes. Furthermore, the *kaiA3B3* cluster is usually extended by *kaiC3*, which resembles the structure of the canonical *kaiABC* gene cluster, with only two exceptions (*Synechocystis* and *Microcystis aeruginosa* NIES-843) (Fig. 1b). Interestingly, *kaiA3B3C3* synteny was also found in other prokaryotic genomes that harbor orthologs of *kaiA3*, except for *Chloroflexus aggregans* DMS 9485 (Fig. 1b). Furthermore, we detected strong significant co-occurrences between KaiA3 and KaiB3 ($p < 0.0001$) as well as between KaiA3 and KaiC3 ($p < 0.0001$; Fig. 1c) in organisms encoding KaiC1. The co-occurrence of KaiB3 and KaiC3 has been previously shown[27]. Thus, KaiA3 forms a distinct set of proteins with KaiB3 and KaiC3, which showed no further significant co-occurrence with other clock components (Fig. 1c[27]). Altogether, both datasets suggest a functional relationship between KaiA3 and the KaiB3-KaiC3 system.

## KaiA3 interacts with and promotes autokinase activity of KaiC3

Using yeast two-hybrid (YTH) experiments, we verified the interaction between the clock proteins KaiC3 and KaiA3 (Fig. 2a, Fig. S5), which is consistent with a previous large-scale protein-protein interaction analysis by Sato et al.[39]. Although KaiA3 clearly interacted with KaiC3, an interaction with KaiB3, the second element of the KaiB3-KaiC3 clock

system, was not detected (Fig. S5b). This is not surprising, as it has been demonstrated that the interaction between the *Synechococcus* proteins KaiA and KaiB requires the presence of KaiC[47]. To further characterize the interaction of the proteins in vitro, we heterologously expressed different Kai proteins in *E. coli* and analyzed complex formation using clear-native PAGE (Fig. 2b and Fig. S6). The His-tagged KaiA3 protein (monomer: 35 kDa) migrated as a single band of approximately 100 kDa in size, indicating the formation of KaiA3 homo-oligomers, at least dimers. *Synechococcus* KaiA migrated at -60 kDa, in line with previous results[48], confirming the formation of KaiA dimers. The discrepancy in the migration pattern between KaiA3 (His-tagged) and KaiA (GST-tag removed) might be due to differences in their predicted charge (−19.17 for KaiA and −7.94 for KaiA3, respectively, at pH 7.0). Recombinant KaiB3 (monomer: 12 kDa) was shown to form monomers and tetramers after size exclusion chromatography[26]. KaiB3 displayed three distinct bands in the native gels (Fig. 2b). The two lower bands most likely represent the monomeric and tetrameric forms, whereas the uppermost band (-67 kDa) could be an impurity in the protein preparation[26]. Recombinant KaiC3 was produced with an N-terminal Strep-tag[26]. Strep-tagged KaiC3 (monomer: 58 kDa) migrated as one band between 272 and 450 kDa and could represent a hexameric complex (348 kDa). Incubation of KaiC3 with KaiA3 alone led to protein accumulation in the wells in native PAGE, indicating precipitation of the KaiA3/KaiC3 complex in the absence of KaiB3 (Fig. 2b). However, the interaction between KaiA3 and KaiC3 was validated by immunoprecipitation-coupled liquid chromatography-mass spectrometry (LC-MS) analysis of FLAG-tagged KaiC3 (Fig. S7). Furthermore, the experiments did not reveal any interactions between KaiA3 and either KaiC1 or KaiC2 (Fig. S5, Fig. S7), indicating the specificity of the KaiA3-KaiC3 interaction. No complex formation was detected between KaiA3 and KaiB3 (Fig. 2b, Fig. S5, Fig. S6). In contrast, the formation of a large protein complex was observed when all three clock components, KaiA3, KaiB3, and KaiC3, were incubated together for 16 h at 30 °C (Fig. 2b; Fig. S6). The size matches that of a complex consisting of one KaiC3 hexamer, six KaiA3 dimers, and six KaiB3 monomers (840 kDa). The presence of KaiA3 in the complex was validated by western blot analysis using an anti-His antibody (Fig. 2b, Fig. S6). As expected, no such complex was formed when KaiA3 was replaced by *Synechococcus* KaiA (Fig. 2b). Moreover, no such complex was formed when KaiB3 was replaced by its isoform KaiB1, suggesting that KaiB3 is specific for KaiA3 as well and that KaiB3 might recruit KaiA3 to the KaiC3/KaiB3 complex (Fig. S6).

Previous studies have shown that KaiC3 has autokinase activity that is independent of KaiA1[26,28]. Since our studies revealed an interaction between KaiC3 and KaiA3, we were interested in probing the influence of KaiA3 on the phosphorylation of KaiC3. The recombinant Kai proteins described above were used for this purpose. KaiC3 was incubated for 16 h at 30 °C in the presence or absence of other Kai proteins, and its phosphorylation state was analyzed by LowC-SDS-PAGE (Fig. 2c), and LC-MS/MS (Fig. S8). Because KaiC3 was partially phosphorylated after purification from *E. coli*, the protein preparation was incubated for 18 h at 30 °C prior to the start of the assays. During this incubation period, KaiC3 was autodephosphorylated, as is typical for KaiC proteins (Fig. 2c, NP-KaiC3)[45]. The addition of KaiA3 led to the phosphorylation of KaiC3, while the presence of *Synechococcus* KaiA had no influence on the phosphorylation state of KaiC3. In contrast, KaiC3 dephosphorylation was enhanced by KaiB3 (Fig. 2c, upper panel). Replacing KaiB3 with its isoform KaiB1 in samples containing KaiA3 maintained KaiC3 in the phosphorylated state (Fig. 2c, lower panel). Analysis of KaiC3 phosphorylation by LC-MS/MS- identified the neighboring residues Ser423 and Thr424 as phosphorylation sites, which are conserved across KaiC1 and KaiC3 homologs (Fig. S8). Based on these analyses, we conclude that KaiA3 likely has a KaiA-like function in promoting the phosphorylation of KaiC3 and interaction with KaiB3, which in turn enhances dephosphorylation. Neither

*Synechococcus* KaiA nor *Synechocystis* KaiB1 could substitute for KaiA3 or KaiB3, respectively, demonstrating that the *Synechocystis* KaiA3/KaiB3/KaiC3 proteins represent a functional complex.

## KaiC3 phosphorylation cycles in vitro and in *Synechocystis* cells

The opposing effects of KaiA3 and KaiB3 on KaiC3 phosphorylation imply that these three *Synechocystis* proteins may form a functional in vitro oscillator. When *Synechococcus* KaiABC proteins were mixed in vitro, the KaiC phosphorylation rhythms were relatively insensitive to KaiB concentration but could occur only in a small window of KaiC:KaiA ratios. In the presence of KaiB, low KaiA concentrations were insufficient to increase KaiC phosphorylation, whereas excessive KaiA could not be counteracted by KaiB[49]. We observed the same effect for the *Synechocystis* proteins, when we incubated a constant KaiC3:KaiB3 ratio with various KaiA3 concentrations: 0.5 μM KaiA3 failed to change the phosphorylation of KaiC3 within 48 h. 8.4 μM KaiA3 stimulated hyperphosphorylation of KaiC3 within 6 h, and KaiC3 remained highly phosphorylated afterwards (Fig. 3, Fig. S9a). In the presence of 1.4–4.2 μM KaiA3, the KaiC3 protein was also phosphorylated within 6 h and then dephosphorylated until the 18 h time point. Thus, for the three intermediate KaiA3 concentrations, we observed one in vitro cycle of phosphorylation and dephosphorylation. The stimulating effect of KaiA3 on KaiC3 phosphorylation was saturated at a KaiA3 concentration of 4.2 μM, which corresponds to a KaiA3:KaiC3 stoichiometry of 1:0.8. Only in the presence of 1.4 μM and 2.8 μM KaiA3 (corresponding to a -1:1.4 and 1:2.2 stoichiometry of KaiA3:KaiC3), we could reconstitute another weak phosphorylation cycle peaking at 30 h (Fig. 3c).

The KaiC3 phosphorylation cycles at intermediate KaiA3 concentrations fulfilled one of the defining criteria of a true circadian rhythm: they displayed a -24 h period. They also continued for more than one cycle. However, the oscillation displayed a low amplitude and appeared dampened. Therefore, we tested another criterion of true circadian rhythm, which is temperature compensation. A mixture of KaiC3, KaiB3, and KaiA3 was incubated at different temperatures. We did not detect a period change between 30 °C and 35 °C. At 25 °C, the period was extended to approximately 30 h, indicating temperature compensation to a similar extent as that observed for the *Synechococcus* KaiABC oscillator[1]. In this experiment, we extended the analysis at 30 °C to 60 h and observed that the dampened oscillation continued.

These observations led us to investigate the potential persistence of KaiC3-phosphorylation rhythms in *Synechocystis* across successive free-running cycles. We entrained wild-type *Synechocystis* using two 12 h LD cycles. After release to constant light (LL), we collected samples over a course of three days and separated total proteins via SDS-PAGE for western blot analysis. Probing with a specific antibody (Fig. S9b) revealed stable, high-amplitude oscillations in KaiC3 phosphorylation (Fig. 4). The oscillations displayed a -24 h period and persisted for (at least) three free-running cycles in the cellular context (Fig. 4). The timing of hypo- and hyperphosphorylation was comparable to that reported for *Synechococcus* KaiC[6,50,51]. KaiC3 was phosphorylated towards the end of the subjective day and dephosphorylated around the subjective dawn. In addition, the abundance of KaiC3 oscillated over a period of approximately 24 h (Fig. 4), as published for *Synechococcus* KaiC[6,11,50,52].

## KaiC1 and KaiC3 phosphorylation are phase-locked in *Synechocystis* cells

Our in vitro and in vivo phosphorylation data indicated that the *Synechocystis* KaiA3B3C3 system functions similarly to the well-studied KaiABC oscillator in *Synechococcus*. Initially, KaiA1B1C1 proteins were predicted to form this ortholog based on their sequences and interactions, as reported in previous studies[26,28]. However, it appears that the KaiA3B3C3 complex may serve as a functional

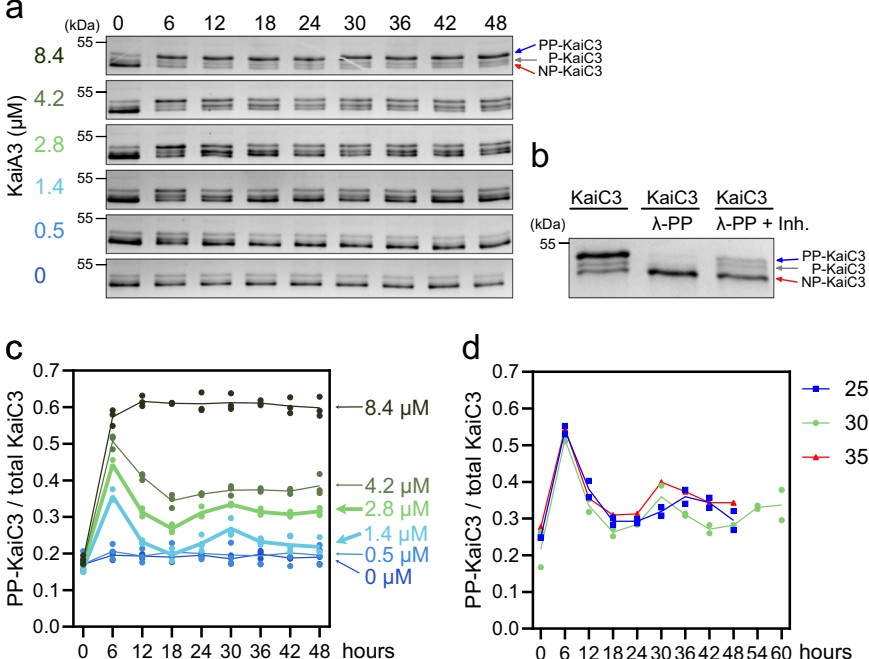

**Fig. 3 | Analysis of KaiA3-dependent KaiC3 phosphorylation in vitro.** KaiC3 (3.4 μM) was incubated with KaiB3 (7.4 μM) and various concentrations of KaiA3 at 30 °C over a time course of 48 h. Aliquots incubated for the indicated time periods were applied to a high-resolution LowC SDS-PAGE gel and proteins were separated in Tris-glycine buffer. **a** Representative gels. **b** To assign the bands to the fully phosphorylated (PP-KaiC3), single-phosphorylated (P-KaiC3) and non-phosphorylated (NP-KaiC3) forms, KaiC3 was dephosphorylated with Lambda phosphatase (KaiC3/λ-PP) (n = 3) in the presence or absence of phosphatase inhibitors (PhosStop (n = 1) and vanadate (n = 2), KaiC3/λ-PP +Inh). **c** The ratio of PP- KaiC3 to total KaiC3 based on gel images (representative gels are shown in (**a**)). Each line/color represents a different KaiA3 concentration. The lines show the average from three assays (points). **d** Representative assay to test temperature compensation. KaiC3 (3.4 μM), KaiB3 (7.4 μM) and KaiA3 (2.8 μM) were mixed, aliquots were incubated at 25 (blue), 30 (green) or 35 °C(red) for the indicated times (n = 2 for all temperatures, except 35 °C (n = 1)) and the ratio of PP-KaiC3/total KaiC3 was determined as described above. Lines indicate the mean. Source data are provided as a Source Data file.

equivalent to the *Synechococcus* KaiABC oscillator. Alternatively, it is possible that the two KaiABC systems coexist and are interconnected within *Synechocystis*. To answer these questions, we analyzed proteins from the above-described experiment, covering three free-running days after LD synchronization (Fig. 4), using an antibody specific for KaiC1 (Fig. S9b). KaiC1 phosphorylation displayed stable ~24 h rhythms which were phase-locked with the KaiC3 phosphorylation rhythm. However, the amplitude was reduced compared with that of KaiC3 phosphorylation, and KaiC1 abundance showed only weak changes.

**The two KaiABC systems together drive circadian backscatter rhythms**

The observed in vivo phosphorylation rhythms of KaiC1 (Fig. 4) implied that KaiA1B1C1 also forms a functional oscillator in *Synechocystis*. Oscillators can function as independent systems that drive separate rhythmic outputs, or only one system is a bona fide oscillator in the cell that controls the rhythmicity of the other. Alternatively, the oscillators may be dependent on each other, form only one complex, or are at least interconnected and integrated into one circadian output. Phase locking of the in vivo phosphorylation rhythms of KaiC1 and KaiC3 (Fig. 4), together with the observation that components of the two oscillators interact with each other in the cell[26], support the latter hypothesis. Quantifying discrete outputs is challenging because of the yet-to-be-identified nature of the output components in the KaiA3B3C3 system. Therefore, we aimed to monitor a more general circadian rhythm that can be detected by backscatter measurements during growth in liquid cultures[53]. Even without knowing the distinct output of each oscillator, this enabled us to test the hypothesis that the two oscillators operate together.

The backscatter properties of *Synechocystis* cells oscillate with a ~24 h period under LL, after cultures are synchronized by dilution with fresh medium. These circadian oscillations are temperature-compensated and driven by the *kaiA1B1C1* gene cluster[53]. To understand whether this output can be used as a general readout of the circadian status of the cell, we investigated the effect of *ΔkaiA3B3C3* deletion on backscatter oscillations in comparison with wild-type *Synechocystis* and *ΔkaiA1B1C1* strains. We grew the three strains as two subsequent pre-cultures for ten days in LL, synchronized them by dilution to OD$_{750nm}$ of 0.9, and monitored the backscatter over time. The presence and loss of oscillations were already apparent in the raw backscatter data (ref. 53 and Fig. 5a–c). For a better presentation, we subtracted a polynomial regression fit from the raw backscatter signal to remove the influence of culture growth, and subtracted the average difference (Fig. 5d–h and Fig. S10). Deletion of the whole *kaiA3B3C3* system led to an intermediate circadian output when compared to *Synechocystis* wild type and *ΔkaiA1B1C1*. Backscatter oscillations of the wild type could be described by a simple harmonic oscillation with a ~26 h period (26.46 ± 0.34 h, n = 3) (Fig. 5d, Fig S10a, b). In the *kaiA1B1C1* deletion strain, the oscillation was almost abolished, as reported by Berwanger et al.[53]. Upon close inspection, however, we observed extremely low amplitude oscillations, which were best described by a simple harmonic oscillation with a ~33 h period (33.32 ± 3.57 h, n = 3) (Fig. 5e, Fig S10a, b). Deletion of the *kaiA3B3C3* system resulted in reduced and potentially dampened oscillations, which were not well described by the simple harmonic cosine function (dotted line in Fig. 5f). The amplitude and period changed over the course of the experiment. To allow comparison with the wild type, we determined the amplitude and phase of the first backscattering peak as well as the period of the first backscattering cycle. We detected a phase

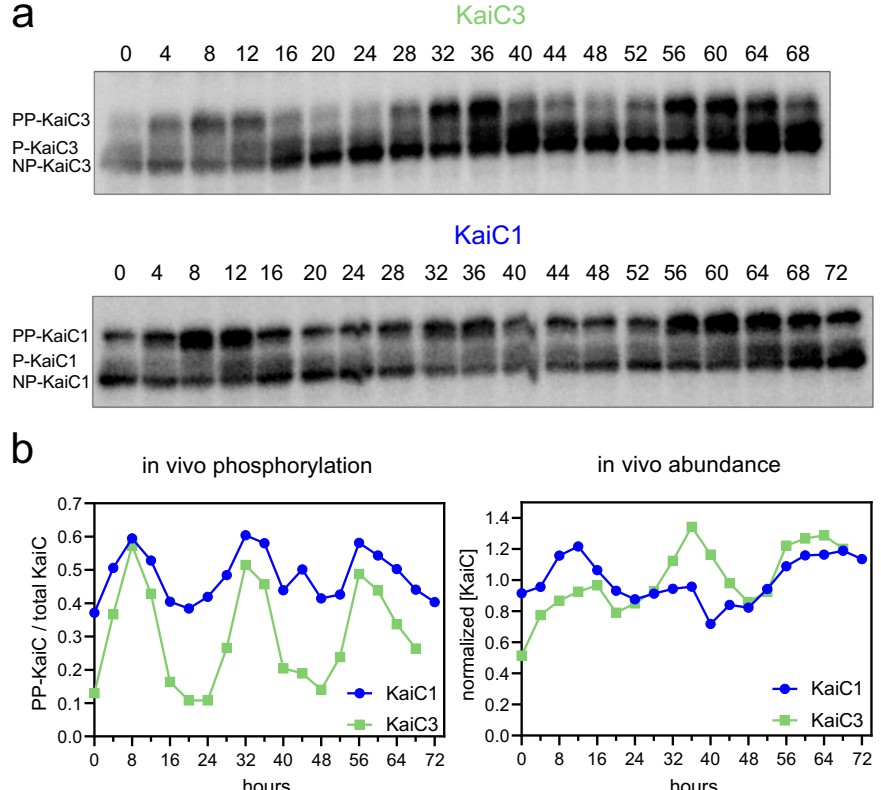

**Fig. 4 | KaiC3 and KaiC1 protein abundance and phosphorylation profiles in *Synechocystis* cells grown under continuous illumination conditions after synchronization. a** Western blots showing KaiC3 and KaiC1 phosphorylation over three days. The cells were cultured in liquid medium, entrained with two LD cycles, and then returned to LL. Cells were harvested every 4 h at LL. 8 µg of total protein were loaded in each lane and subjected to western blot analysis. We designated the upper bands as hyperphosphorylated (PP-KaiC) and the lower bands as hypophosphorylated (P- and NP-KaiC), analogous to our in vitro data and the *Synechococcus* KaiC, and based on Fig. S9b. The experiment was performed once over three LL cycles and confirmed once over one LL cycle. **b** Densitometric estimation of rhythms in the KaiC3 and KaiC1 fractions of PP-KaiC (left) and KaiC protein abundance (right) from the blots shown in (**a**). Source data are provided as a Source Data file.

shift of ~-7 h and a reduction in the amplitude to one-third in comparison to the first backscattering peak in the wild type (Fig. 5i). Furthermore, the period of the first backscattering cycle was significantly shortened by ~5 h in comparison to the wild type (Fig. 5j, Fig. S10c). Altogether, the backscattering data imply that the KaiA1B1C1 system mainly drives circadian rhythms, but requires KaiA3B3C3 to maintain the period and amplitude. On the other hand, KaiA3B3C3 may be able to drive low-amplitude oscillations but requires coupling to KaiA1B1C1 to maintain a circadian period and ensure a high amplitude.

To dissect the role of the single components of the new KaiA3B3C3 system, we monitored the effects of single and double mutants of *kaiA3*, *kaiB3*, and/or *kaiC3* on backscatter oscillations. All strains in which *kaiC3* was deleted, alone or in combination with other genes, displayed the same phenotype as *Synechocystis ΔkaiA3B3C3* (Fig. 5g, i, j; Fig. S10a), confirming that KaiC3 was the central protein in this system. Deletion of either *kaiA3* or *kaiB3* had different effects (Fig. 5h–j; Fig. S10a). Knockout of *kaiA3* led to a similar decrease in amplitude as observed in *ΔkaiC3*. However, the phase shift of the first peak of the dampened oscillation was less pronounced than that of *ΔkaiC3*, and the period of the first low-amplitude cycle was only slightly lower than that of the wild type (Fig. 5i, j, Fig. S10c). A double knockout of *kaiA3* and *kaiB3* resulted in a phenotype similar to that of the deletion of *kaiA3* alone (Fig. 5h–j, Fig. S10). In contrast, the deletion of *kaiB3* almost completely abolished these oscillations. In general, a high variance among the experiments was observed for this mutant (Fig. 5h, i; Fig. S10a). On average, single *kaiB3* deletion drastically reduced the amplitude. Notably, when plotting the relative amplitude

against the phase shift of the first peak, this mutant was grouped with the *ΔkaiA1B1C1* mutant rather than with strains carrying mutations in the genes encoding the components of the KaiA3B3C3 system (Fig. 5i). Accordingly, the low-amplitude oscillations fitted well to harmonic oscillations without dampening (Fig. S10a) as was observed for the *kaiA1B1C1* strain. The period of the simple harmonic oscillation fit to the backscattering oscillations in the *ΔkaiB3* strain (39.04 ± 2.58 h) was extended in comparison to the wild type and *ΔkaiA1B1C1* mutant (Fig. S10b).

Overall, this suggests that KaiC1 drives stable free-running oscillations, and is the core of the main system that drives the backscatter rhythms of the cell. However, backscatter rhythms persist only as self-sustained oscillations if a second post-translational putative oscillator, KaiA3B3C3, is also present, which implies that the two systems are directly or indirectly connected.

## Mutation of kaiA3 impacts growth and viability during mixotrophic and chemoheterotrophic growth

Previously, we showed that deletion of the *kaiA1B1C1* operon severely affects the viability of cells on agar plates[30]. When grown photoautotrophically in LL, the *ΔkaiA1B1C1* mutant strain behaved like the wild-type strain. However, *ΔkaiA1B1C1* was not able to grow under chemoheterotrophic conditions, and viability was reduced under mixotrophic conditions (in LL) as well as in LD cycles under both conditions[26,30]. Previously, we also revealed that deletion of *kaiC3* had less detrimental effects: growth was reduced in chemoheterotrophic conditions, but not in LL or LD, independent of the presence or

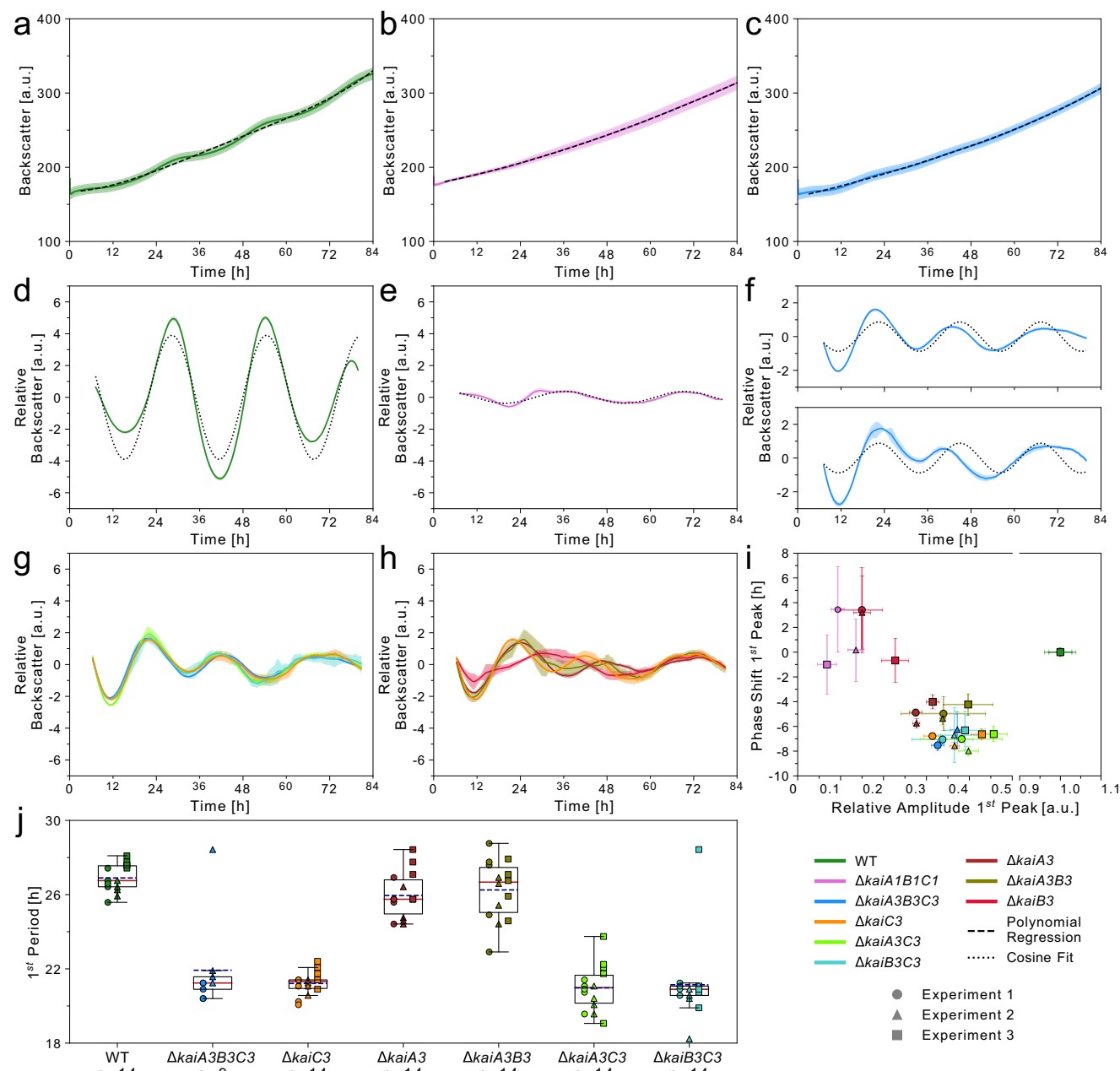

**Fig. 5 | Detection of circadian rhythms using backscatter measurements.**
**a**–**c** Growth of *Synechocystis* wild type (WT) (**a**), *Synechocystis ΔkaiA1B1C1* mutant strain (**b**), and *Synechocystis ΔkaiA3B3C3* mutant strain (**c**) in LL after initial synchronization by dilution. Graphs display the average backscatter signal as a rolling average (solid line) at 730 nm with SD (shaded) from a representative experiment with 4–5 wells per strain. Polynomial regression (dashed line) was fitted to the data. **d**, **e** For each replicate within the experiments with wild type (**d**) and *Synechocystis ΔkaiA1B1C1* (**e**), the raw backscatter signal was subtracted from the polynomial regression fit to remove the contribution of the overall growth of the cultures. The average difference was subtracted for normalization. Displayed is the average of this normalized backscatter (solid lines) with SD (shaded). Curves are smoothed. The dotted line indicates a simple harmonic cosine fit. **f** Same as (**d**, **e**), but the two experiments investigating backscatter in the *ΔkaiA3B3C3* mutant are shown. **g**, **h** Same as (**d**, **e**) for further *Synechocystis kai* gene mutants without the cosine fits. All data displayed in (**d**–**h**) were collected from the same representative experiment (except for the additional *Synechocystis ΔkaiA3B3C3* in **f**). Data from all three independent experiments are shown along with the cosine fit in Fig. S10 (except for *Synechocystis ΔkaiA3B3C3*). **i** To determine the effect on the first cycle after synchronization, the phase shift and relative amplitude of each *Synechocystis* mutant were calculated for the first peak (approx. between 21–29 h) for three independent experiments (Fig. S10a). The circles, triangles, and squares represent the first, second, and third experiments, respectively. *Synechocystis ΔkaiA3B3C3* was not included in the third experiment. Marker scales are proportional to the number of replicate wells within one experiment (Exp. 1: *n* = 5 for all strains except *ΔkaiA1B1C1* (*n* = 4), Exp. 2: all strains *n* = 4, Exp 3: all strains *n* = 5). Error boundaries were calculated using formulas (3) and (5) (see Methods section). The *X*-axis is discontinuous. **j** For strains that displayed dampened backscatter oscillations, we derived the length of the first period from the distance of the first trough to the first peak (see materials and methods) for the three experiments in Fig. S10a (*Synechocystis ΔkaiA3B3C3* only two experiments). The backscattering period of the wild type was determined in the same way for comparison. Boxes are based on all replicate wells (indicated as n) from all three experiments (discriminated by the same symbol) and range from the first to the third quartile. Whiskers extend to the furthest data point within 1.5x the interquartile distance. The red line indicates the median and the dashed line indicates the mean. Results of the pairwise statistical tests are shown in Fig. S10c. Source data are provided as a Source Data file.

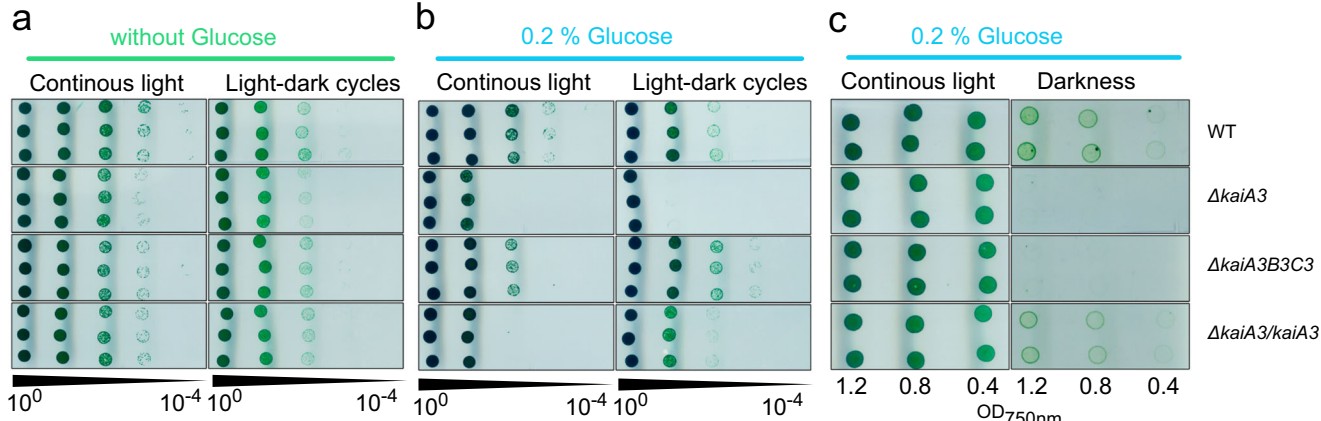

**Fig. 6 | Deletion of *kaiA3* and overaccumulation of *kaiA3*, result in growth defects during mixotrophic and chemoheterotrophic growth. a–c** Proliferation of the wild type (WT), the Δ*kaiA3* and Δ*kaiA3B3C3* deletion mutants, and the Δ*kaiA3/kaiA3* complementation strain under different growth conditions. Strains were grown in liquid culture in LL, and different dilutions were spotted on agar plates and incubated under the indicated light conditions, with a light phase corresponding to 75 μmol photons m$^{-2}$ s$^{-1}$ white light. Representative result from three independent experiments are shown. **a** Cultures were diluted to an OD$_{750nm}$ value of 0.4, and tenfold dilution series were spotted on agar plates. Plates were analyzed after 6 or 8 d of LL and 12 h LD cycles, respectively (photoautotrophic growth). **b** Same as (**a**), but the cells were spotted on agar plates containing 0.2% glucose (photomixotrophic growth). **c** Cultures were diluted to OD$_{750nm}$ values of 1.2, 0.8, and 0.4, and spotted on agar plates supplemented with 0.2% glucose. The plates were analyzed after 26 d of continuous darkness (chemoheterotrophic growth). Because of the higher cell density, a second plate was prepared and incubated in LL for three days (photomixotrophic growth). Source data are provided as a Source Data file.

absence of glucose[26]. To further determine whether the two systems control similar physiological functions and, therefore, might be connected, we performed viability assays with mutants affecting the *kaiA3* gene. Therefore, the Δ*kaiA3B3C3* and Δ*kaiA3* mutant strains, as well as the genomic Δ*kaiA3/kaiA3* complementation strain, were analyzed under various growth conditions. These analyses were performed in the *Synechocystis* PCC-M wild-type background strain, which has been used in previous studies on the KaiA1B1C1 system and is known to grow in complete darkness[28,30]. Furthermore, this strain is motile and aggregates in liquid culture. Therefore, we performed spot assays and did not measure growth in liquid cultures. The cell suspensions were plated on agar at different dilutions and grown photoautotrophically (Fig. 6a) and photomixotrophically (Fig. 6b) in LL and 12 h LD cycles or chemoheterotrophically (Fig. 6c). Because the strains grew very slowly under chemoheterotrophic conditions, the cells were spotted at higher concentrations under these conditions. The mutant strain lacking *kaiA3* and the triple-knockout strain showed a phenotype similar to that previously observed for a *kaiC3* deletion mutant. There were almost no differences in the viability of the mutant strains compared to that of the wild type under photoautotrophic conditions under LL and LD cycles (Fig. 6a). However, they were unable to grow in the dark (Fig. 6c). This ability was fully restored when *kaiA3* was reinserted into the *kaiA3* deletion strain (Fig. 6c).

Under photomixotrophic conditions, the mutant strain lacking all three alternative *kai* genes (Δ*kaiA3B3C3*) exhibited a growth phenotype similar to that previously observed for Δ*kaiC3*[30]. The strain proliferated well and, in LD cycles, seemed to have some advantages compared to the wild type (Fig. 6b). However, the Δ*kaiA3* strain showed less viability under photomixotrophic conditions, a phenotype comparable to that of the Δ*kaiA1B1C1* strain (Fig. 6b and ref. 30). Again, viability was partly restored by re-insertion of *kaiA3* (Fig. 6b).

Surprisingly, overexpression of *kaiA3* by insertion of a KaiA3-FLAG encoding plasmid in the wild-type background reduced viability to almost the same extent as *kaiA3* deletion (Fig. S11). To exclude the possibility that this phenotype was caused by the FLAG-tag or expression from the plasmid, we inserted the same plasmid into the Δ*kaiA3* deletion strain. The FLAG-tagged KaiA3 complemented the *kaiA3* deletion in the same manner as genomic complementation with non-tagged KaiA3 (Fig. S11). These findings suggest that deregulation

of the phosphorylation level of KaiC3 might affect the viability of *Synechocystis* under photomixotrophic and chemoheterotrophic conditions more than the deletion of *kaiC3* or the whole KaiA3B3C3 system. In the absence of KaiA3 and the presence of excess KaiA3, when, according to the in vitro data, KaiC3 is constantly hypo- or hyperphosphorylated, respectively (Fig. 3c), the phenotypes are as detrimental as for the knockout of *kaiA1B1C1*. To confirm that KaiC3 phosphorylation is indeed dependent on KaiA3 levels in the respective mutants, we grew *Synechocystis* cells in an LD cycle, followed by constant illumination, separated whole-cell extracts on a Phos-tag gel, and identified KaiC3 phosphorylation forms by western blot analysis (Fig. 7a–c). Two or more bands were detected in vitro and in vivo, which partly overlapped with a non-specific band detected in the Δ*kaiC3* strain (Fig. 7a). These bands might reflect single phosphorylated states of KaiC3, but were not included for densiometric quantification of KaiC3 phosphorylation to exclude the effects of potential cross-reactions of the antibody with KaiC1, KaiC2, or another protein (Fig. 7d). In the wild type, phosphorylation cycled with a period of about 24 h. In contrast, KaiC3 was mostly dephosphorylated in the Δ*kaiA3* mutant strain. There is still some fully phosphorylated KaiC3 detectable in this mutant, which might originate from weak autophosphorylation in the absence of KaiA[28]. In the KaiA3 overexpression strain, KaiC3 was highly phosphorylated compared to the wild type. Phosphorylation after *kaiA3* overexpression varied strongly between the experiments, most likely due to different KaiA3 levels when using the copper-dependent P$_{petJ}$ promoter for overexpression. Thus, the phenotypes of the different mutants suggest that both clock systems are involved in regulating heterotrophic growth under light and darkness. The interconnection and (putative) role of the components in metabolic control and generation of backscatter oscillations are summarized in the model in Fig. 8.

## Discussion

Our knowledge of the function, composition, and network of clock systems in prokaryotes, including cyanobacteria, is steadily increasing. Even though multiple copies of the core clock proteins KaiB and KaiC are encoded in bacterial genomes, the canonical KaiA was found only as a single copy in Cyanobacteria yet[27,28,43,54]. By identifying a chimeric KaiA3 and verifying its interaction with the KaiB3-KaiC3

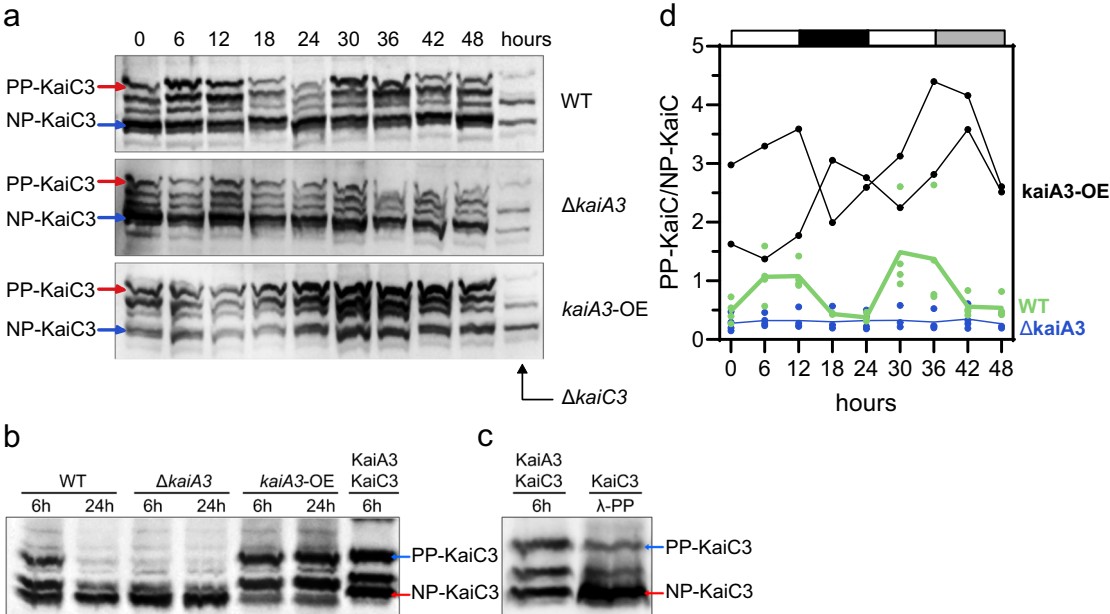

**Fig. 7 | Phosphorylation of KaiC3 in *Synechocystis* (PCC-M) wild-type, *kaiA3* mutant (Δ*kaiA3*), and *kaiA3* overexpression (*kaiA3-OE*) strains grown under photoautotrophic conditions.** Samples were collected every 6 h from cells grown in a 12 h LD cycle, followed by LL. Whole-cell extracts were separated using Phos-tag SDS-PAGE and immunodecorated with a KaiC3-specific antiserum. Representative blots are shown in (**a**). We detected 4-5 bands which partially overlapped or were slightly shifted compared to the bands detected in the Δ*kaiC3* strain (12 h time point was loaded). The two indicated prominent bands were absent in the Δ*kaiC3* strain. **b** Whole cell extracts of *Synechocystis* wild-type (WT), *kaiA3* mutant (Δ*kaiA3*), and the overexpression (*kaiA3*-OE) strain grown for 6 or 24 h in a 12 h LD cycle were loaded together with in vitro phosphorylated recombinant KaiC3 (KaiA3-KaiC3/

6 h), which was generated by 6 h incubation with 4.2 μM KaiA3 at 30 °C. **c** KaiC3 was dephosphorylated (KaiC3/λ-PP) using Lambda phosphatase and analyzed alongside with in vitro phosphorylated KaiC3 (KaiA3-KaiC3/6 h). Based on the band pattern of recombinant KaiC3, we assigned the indicated bands to fully phosphorylated (PP-KaiC3) and non-phosphorylated KaiC3 (NP-KaiC3). The control reactions in (**b**) and (**c**) were performed at least twice on different gels. **d** The ratio of PP-KaiC3 to NP-KaiC measured in wild-type and Δ*kaiA3* mutant is plotted as average (line) with dots indicating biological replicates (*n* = 4). Data derived from two biological replicates of *kaiA3-OE* are presented as individual curves. The white and dark gray boxes represent the light and dark periods, respectively, and the light gray box represents the subjective night. Source data are provided as a Source Data file.

complex, we added another component to the diversity of bacterial clock systems.

New putative KaiA orthologs have been bioinformatically identified in prokaryotes other than Cyanobacteria[43,55]. Therefore, we suggest that such proteins may play a previously overlooked role in KaiB-KaiC-based systems. Exploring this possibility could provide valuable insights into unanswered research questions, such as the mechanism responsible for the rhythmic processes observed in *Rhodospirillum rubrum*. Indeed, this purple bacterium lacks KaiB1 and KaiC1 orthologs, but possesses KaiA3, KaiB3, and KaiC3 (ref. 56 and Fig. 1). Notably, the recently described primordial oscillator from *Rhodobacter*, which consists of homologs of KaiC2 and KaiB2, can form an hourglass timer without KaiA[22]. A similar primordial clock has been suggested to be present in *Rhodopseudomonas palustris*[21] and the cyanobacterium *Prochlorococcus* MED4[31,32], while other bacterial KaiB and KaiC homologs, including the KaiC2-KaiB2 system from *Synechocystis*, are believed to have clock-independent functions[17,57,58].

It has been proposed that *kaiC* is the oldest evolutionary member of circadian clock genes[54]. KaiC homologs can be found even in Archaea where it was found to control e. g. motility of *Sulfolobus acidocaldarius* by protein interaction[59]. The later addition of KaiB was enough to form a primordial timekeeper which needs a signal for daily resetting of the clock[21,22,31,32]. In *Rhodobacter* KaiC2, dephosphorylation is regulated by the stability of coiled-coil interactions between two connected hexamers as well as by KaiB[22]. Whether autophosphorylation or dephosphorylation dominates depends primarily on the ATP/ADP ratio. Hence, the KaiC2-KaiB2 timer cannot oscillate autonomously but responds to changing ATP/ADP levels. Therefore, it was suggested that the *Rhodobacter* clock represents an ancient timer that depends on changes in photosynthetic activity during the day-night switch[22].

With the evolution of KaiA, a self-sustained oscillator was developed that allowed for true circadian oscillations in gene expression, which can be observed in cyanobacteria. Why does KaiC require KaiA to drive persistent oscillations? By default, the A-loops of *Synechococcus* KaiC hexamers adopt a buried conformation, which inhibits autophosphorylation. Only the binding of KaiA favors phosphorylation by stabilizing A-loop exposure[5]. In contrast, *Rhodobacter* KaiC2 constantly exposes its A-loops, sterically allowing high intrinsic phosphorylation[22]. Furthermore, introducing KaiA as a factor stimulating autophosphorylation of KaiC allows coupling between different KaiC molecules, e.g. by KaiA sequestration, which is needed for synchrony and thus high-amplitude oscillation[13,48,60–62].

The interacting residues between KaiA and KaiC are less conserved in both *Synechocystis* KaiA3 and KaiC3[28,63] (Fig. 1). Since we demonstrated an interaction between KaiC3 and KaiA3, it is likely that co-evolution of the two proteins occurred. KaiC3 does not display the extended C-terminus that contributes to homododecamer formation in *Rhodobacter* KaiC2[22,28], and we only observed the formation of hexamers or smaller oligomers[26] (Fig. 2).

KaiA3 formed a distinct clade at the basis of the KaiA clade. Apart from its presence in the N-terminal domain of the phosphatase RsbU of *Bacillus subtilis*, a distinctive structure of the KaiA C-terminus has rarely been observed[46]. RsbU acts as a positive regulator of the alternative sigma factor B, which is involved in the general stress response[64]. The N-terminal domain of RsbU forms dimers similar to KaiA, and the proposed binding site for its corresponding activator, RsbT, is in an equivalent location to the KaiC-binding site on KaiA[46]. These findings may reflect how protein domains change during evolution, while their original functions are conserved. However, a link between RsbU and the recently proposed circadian clock in *Bacillus*

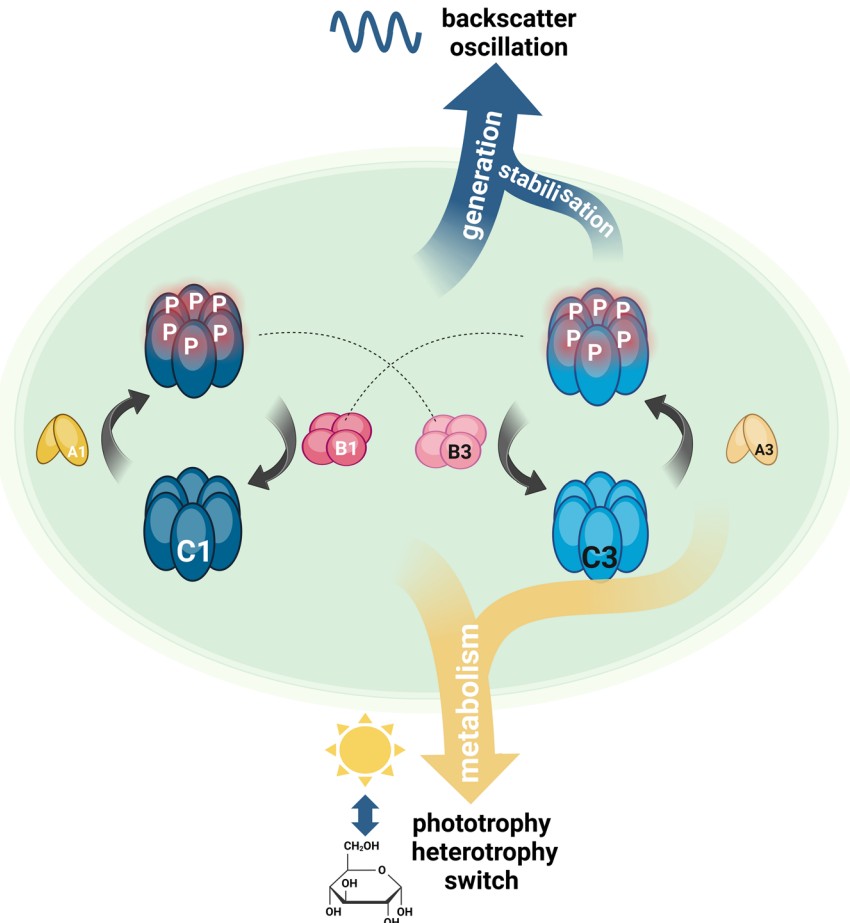

**Fig. 8 | Model of the two interconnected KaiABC systems in *Synechocystis* cells.** KaiC1 and KaiC3 display phase-locked phosphorylation rhythms. KaiA3 and KaiB3 regulate auto-phosphorylation and dephosphorylation of KaiC3 and form a second oscillator. The KaiA1B1C1 oscillator appears to be the main driver of backscatter rhythms, whereas KaiC3 and KaiA3 are required to maintain the amplitude and period. *KaiB3* deletion abolished the rhythms to a similar extent as *kaiA1B1C1* deletion. Because KaiB proteins were shown to also interact with the KaiC proteins of the other system[26], we assume that the absence of competing KaiB3 leads to enhanced KaiB1 binding to phosphorylated KaiC3 (dashed line), thereby disturbing KaiA1B1C1-based oscillations. Altogether, this implies that phosphorylated KaiC3 stabilizes backscatter oscillations. The interconnection between the KaiC1 and KaiC3 systems is also metabolically relevant. Deletion of *kaiA1B1C1* and both up- and downregulation of *kaiA3* reduced mixotrophic growth, whereas deletion of the entire KaiA3B3C3 system and *kaiC3* had no effect. This implies that KaiA1B1C1 mainly contributes to the switch from autotrophic to heterotrophic growth, but the phosphorylation rhythms of KaiC3 can interfere with it. Whether this interference occurs directly via the KaiC1-KaiC3 interaction[26] or indirectly via output pathways needs to be clarified. This figure was created with BioRender.com, released under a Creative Commons Attribution-NonCommercial-NoDerivs 4.0 International license.

*subtilis* has not yet been identified[65]. Moreover, circadian rhythms have been observed in several prokaryotes that do not encode Kai orthologs, suggesting convergent evolution of circadian rhythms in prokaryotes[65,66]. Further in-depth analyses are needed to elucidate whether KaiA3, together with KaiB3 and KaiC3, or the well-studied *Synechococcus* circadian clock present a more ancestral system, because analysis of a larger dataset recently suggested that the canonical *kaiA* gene evolved at the same time as cyanobacteria[43].

In this work, we broadly define an oscillator to include systems that may be dampened but nevertheless have a natural frequency. Taken together, our data are consistent with a model in which KaiA3 fulfills the functions of a true KaiA homolog, such as dimerization, binding to KaiC3, and enhancing KaiC3 autophosphorylation. Other mechanistic processes, such as sequestration to the CI ring by binding to KaiB3, remain to be investigated but are clearly possible. By mixing KaiA3, KaiB3, and KaiC3, we reconstituted a dampened in vitro oscillator (Fig. 3), suggesting that the observed in vivo rhythm of KaiC3 phosphorylation is driven by KaiA3 and KaiB3, and that the amount of KaiA3 is critical for the phosphorylation rhythm.

In *Synechococcus* KaiC, ATPase activity directly correlates with the clock period and mediates temperature compensation[4]. We observed temperature compensation of dampened KaiC3 phosphorylation in the presence of KaiA3 and KaiB3, although the ATPase activity of KaiC3 alone is temperature dependent[26]. Future work might reveal how the presence of KaiA3 and KaiB3 contributes to temperature compensation.

The in vivo phosphorylation of KaiC3 displayed a higher amplitude than the in vitro oscillation, implying that rhythms might be stabilized by other mechanisms in the cell. Whether direct crosstalk between the KaiA1B1C1 and KaiA3B3C3 systems contributes to stabilization remains to be investigated. We can only speculate on the nature of a direct interconnection. KaiC3 is the central protein of the newly identified in vitro oscillator; however, its absence has less severe consequences for backscatter oscillations than the absence of KaiB3. Removing KaiA3 together with KaiB3 restored the dampened oscillation. This suggests that hyperphosphorylated KaiC3 interferes with KaiA1B1C1 driven backscatter oscillations, which could occur directly via sequestration of KaiB1 (see model in Fig. 8).

The *Rhodobacter* hourglass-like timer required environmental cues for daily resetting. However, entrainment by metabolites has also been described for more elaborate and true circadian oscillators. In addition to entrainment by the input kinase CikA[67], the *Synechococcus*

clock can be entrained directly by the ATP/ADP ratio and oxidized quinones[36,68]. Moreover, CikA does not sense light directly, but perceives the redox state of the plastoquinone pool[69,70]. Furthermore, glucose feeding can entrain *Synechococcus* when engineered to take up glucose[71]. In plants, it has been demonstrated that both exogenous sugars and internal sugar rhythms resulting from cyclic photosynthetic activity entrain the clock[72]. *Synechocystis* can naturally utilize glucose, which may make it even more susceptible to metabolic entrainment by sugars. In addition, the need for metabolic compensation[73] may be particularly pronounced. Notably, the *Synechocystis* PCC-M wild-type strain could grow in complete darkness when supplemented with glucose. This is different from an earlier study that showed that *Synechocystis* requires a 5 min blue-light pulse at least once a day to grow heterotrophically in the dark[74]. The authors described this behavior as light-activated heterotrophic growth. There are no studies that explain why cells require this short light pulse, but it is also clear that the PCC-M strain grows fully chemoheterotrophically[30].

In contrast to *Synechococcus*, CikA from *Synechocystis* is a true photoreceptor that binds a chromophore[75]. Thus, it remains unclear whether CikA has a similar function in both cyanobacteria, and whether it interacts with both circadian clock systems in *Synechocystis*. The high structural similarity of the N-terminal domain of KaiA3 to response regulator domains from other organisms indicates that the core structure and activity are maintained, while adaptivity and variation provide specificity for distinct pathways[25]. Within KaiA3, the aspartate residue crucial for phosphorylation is conserved. Theoretically, the protein can receive an input signal from a cognate histidine kinase that has not yet been identified. Thus, there are potentially important differences related to input and output factors, and possibly entrainment of different cyanobacterial circadian clock systems.

The physiological function of the KaiA3B3C3 clock system seems to be related to the different metabolic modes of *Synechocystis*. Mutants deficient in *kaiA3* lose the ability to grow chemoheterotrophically on glucose, which is an aggravated effect compared to *kaiC3*-deficient mutants that merely show reduced growth rates during heterotrophy[26]. Similarly, in *Synechococcus*, disruption of *kaiA* led to one of the most severe effects on activity loss and was traced back to the unbalanced output signaling of the circadian clock[76]. Over-accumulation of KaiA3 also appeared to disturb the *Synechocystis* system (Fig. 6). Such an effect was also shown for the *Synechococcus* clock system, in which increased KaiA levels promote the hyperphosphorylation of KaiC[6,77], thereby deactivating rhythmic gene expression[78]. Surprisingly, inactivation of the complete KaiA3B3C3 system resulted in a different phenotype. Although growth in darkness on glucose was strongly affected, similar to the single mutants, photomixotrophic growth was slightly better in the *ΔkaiA3B3C3* strain than in the wild type.

It is possible that, in the absence of KaiA3, an altered interaction of the KaiC3 system with the KaiC1 system leads to the aggravated growth defects of *ΔkaiA3*. Constant dephosphorylation of KaiC3 may also change its interactions with the KaiA1B1C1 system. Therefore, when the complete KaiA3B3C3 system is missing, KaiA1B1C1 may be able to compensate for this under certain growth conditions. In *Synechocystis*, *ΔkaiA3*-like phenotypes, such as impaired viability during LD cycles or complete loss of chemoheterotrophic growth on glucose, were also observed for *ΔkaiA1B1C1*, *ΔsasA*, and *ΔrpaA* mutants[30,79]. For *ΔsasA*, it was shown that the mutant strain was able to accumulate glycogen but was unable to utilize the storage compound to grow heterotrophically, probably because of its inability to catabolize glucose[79], whereas *Synechocystis ΔkaiA1B1C1* displays a highly reduced glycogen level[53]. A recent metabolomics study suggested that the growth inhibition of *ΔkaiA1B1C1* and *ΔrpaA* mutants in an LD cycle might be at least partly related to a defect in the inhibition of the RuBisCO enzyme in the dark and increased photorespiration, leading to the accumulation of the potentially toxic

product, 2-phosphoglycolate[80]. This previous study also revealed an enhanced growth defect in *ΔkaiA1B1C1* and *ΔrpaA* mutants under photomixotrophic conditions in LD cycles, similar to the *ΔkaiA3* strain in the current study. This further supports the idea that the KaiA3B3C3 system is interconnected with the core clock system KaiA1B1C1.

Clearly, there is a difference in the phenotypes between our study and the results demonstrated by Zhao et al.[17], who analyzed single and double *kaiB3* and *kaiC3* knockout strains. In LD cycles, the *kaiB3C3* knockout strain showed a reduced growth rate compared to the wild-type control under photoautotrophic conditions. Even in LL, this mutant showed a reduced growth rate and was outcompeted by the wild-type cells in mixed cultures. Photoheterotrophic and heterotrophic conditions were not tested in this study. *Synechocystis* strains used in different laboratories can vary in their genome and phenotypic characteristics, including glucose sensitivity (see for example[29,81]). As the input and output pathways of the newly discovered KaiA3B3C3 system are unknown, it is possible that mutations in different wild-type variants lead to variations in the expression of phenotypic effects in clock mutants. However, oscillations were observed not only in one particular laboratory strain, but also in different *Synechocystis* variants using different equipment (Fig. 4 and Fig.7). Another reason why we used different *Synechocystis* variants in this study was to allow comparison with previous studies. In addition, different laboratory strains were better suited for specific analyses (e.g., recording backscatter signals required non-aggregating strains). In addition, mutations in *kai* genes have similar effects on the backscatter rhythm of the strain investigated here (originating from the University of Uppsala) and on bioluminescence rhythms in the strain reported by Zhao et al.[17] (Vanderbuilt University). In both background strains, *kaiA1B1C1* deletion abolished the recorded oscillations. Furthermore, *kaiC3* deletion dampened backscatter rhythms with a reduced amplitude in the first cycle (Fig. 5). Interestingly, Zhao et al. also revealed a reduced amplitude peak in luminescence rhythms in their *kaiC3* mutant strain[17].

Both studies (ref. 17, this work) suggest that the KaiA1B1C1 system is the master clock in *Synechocystis*, and that the KaiA3B3C3 system provides some redundancy and might stabilize oscillations. The nature of backscattering rhythms is not clearly resolved yet, but was suggested to be related to glycogen metabolism, because glycogen is known to display circadian synthesis and degradation rhythms in *Synechococcus*[68,82,83]. Consistent with this, no backscattering oscillations were detected in a *glgC* mutant, which is defective in glycogen synthesis[53,84]. Alternatively, the oscillations might reflect cell division, which is known to be regulated in a circadian fashion in *Synechococcus*[85–87]. Circadian clock systems that are composed of multiple oscillators are widespread in multicellular eukaryotic organisms[88]. Coupling of oscillators between cells has also been observed in filamentous cyanobacteria, but not between unicellular *Synechococcus* cells[89,90]. Early studies using the dinoflagellate *Gonyaulax polyedra* (renamed *Lingulodinium polyedra*) have suggested that at least two oscillators can exist in a unicellular organism[91,92]. Two detectable circadian rhythms (bioluminescence and aggregation) were relatively independent, with different periods and phase responses under certain conditions, and hence, able to decouple. Therefore, the authors concluded that separate oscillators individually control each rhythm[91–93]. The molecular oscillators have not been identified yet[94]. In *Synechocystis*, the deletion of each single Kai system had different effects on the period of backscatter oscillation (Fig. 5), which implies that two distinct oscillators exist. The KaiA3B3C3 oscillator may drive the long-period oscillation with an extremely low amplitude, which we detected in the *kaiA1B1C1* mutant. The KaiA1B1C1 system might generate a dampened rhythm in the *kaiA3B3C3* deletion strain, which started with a reduced period compared to the wild type. In the wild type, the phosphorylation cycles of KaiC1 and KaiC3 were

synchronized, and a 24 h backscatter rhythm was detected, indicating that both systems operated jointly to regulate the same functions within a coupled system. However, we cannot rule out the possibility that the two systems function separately or have different phases under untested conditions. In eukaryotic organisms, circadian clock systems have been described that are composed of multiple oscillators[88]. This allows oscillators to respond to different environmental signals and to control rhythms in different output paths. The coupling strength of multiple oscillators balances the stability and precision of the timing mechanism with the flexibility of entrainment[95]. For *Lingulodinium polyedra*, it has been suggested that the coupling of the two oscillators provides higher adaptivity to the availability of resources[93]. In particular, the connection between circadian rhythms and metabolism opens up new perspectives in the field[96].

In summary, we demonstrated that KaiA3 is a novel KaiA homolog and an element of the KaiC3-based signaling pathway with canonical KaiA functions. The N-terminal half of KaiA3 may still have a response regulatory function and may connect the whole system to other regulatory elements. KaiA3 must be located within the regulatory and metabolic networks of *Synechocystis*. Finally, our findings demonstrate that *Synechocystis* encodes two KaiABC-based protein oscillators. Both systems are required to drive rhythmicity and ensure the growth of *Synechocystis* with exogenously supplied glucose. Compared to the well-understood *Synechococcus* circadian clock system, the more versatile lifestyle of *Synechocystis* may require a more complex and redundant regulatory network.

## Methods

### Reciprocal BLAST of Sll0485 (KaiA3) and Slr1783 (Rre1)

Reciprocal BLAST analysis was performed as described at https://doi.org/10.17504/protocols.io.q3rdym6 and by Schmelling et al.[27] The 2017 database was used for comparison with existing data on other circadian clock proteins. The protein sequences of Sll0485 (KaiA3) and Slr1783 (Rre1), as references for NarL response regulators[44] from *Synechocystis*, were used as query sequences for this reciprocal BLAST search.

### Co-occurrence analysis

The co-occurrence of KaiA3 with other circadian clock proteins in cyanobacteria containing KaiC1 was examined according to Schmelling et al.[27]. A right-sided Fisher's exact test was used[97]. P-values were corrected for multiple testing after Benjamini-Hochberg[98], with an excepted false discovery rate of $10^{-2}$. All proteins were clustered according to their corrected p-values.

### Synteny analyses using SyntTax

The conservation of gene order was analyzed using the web tool 'SyntTax'[99]. If not mentioned otherwise, default settings (Best match, 10 % norm. BLAST) were applied. Chromosomes were selected manually according to the results of Schmelling et al.[27].

### Multiple sequence alignments with Mafft and Jalview

Sequence alignments, visualization, and analysis were performed with 'Jalview'[100]. The sequences were aligned with Mafft, and if not mentioned otherwise, default settings (L-INS-i, pairwise alignment computation method - localpair using Smith-Waterman algorithm, gap opening penalty: 1.53, gap opening penalty at local pairwise alignment: -2.00, group-to-group gap extension penalty: 0.123, matrix: BLOSUM62) were applied[101]. For analyses of the C-terminus, alignments were trimmed to position 168 in the KaiA reference sequence of *Synechococcus*. After trimming, the alignment was recalculated with Mafft, using the default parameters mentioned above.

### 2D and 3D structure predictions

The alignments generated in Jalview were then used with 'Ali2D' for secondary structure prediction[102] (https://toolkit.tuebingen.mpg.de).

The identity cut-off to invoke a new PSIPRED run was set to 30%. Three-dimensional protein structures were modeled using either Phyre2 or SWISS-MODEL[103,104] (http://www.sbg.bio.ic.ac.uk/phyre2/html/page.cgi?id=index; https://swissmodel.expasy.org/). The resulting structures were analyzed and illustrated using UCSF Chimera[105] (https://www.cgl.ucsf.edu/chimera/).

### Phylogenetic reconstruction of protein trees

Phylogenetic reconstruction of the protein trees of Sll0485 (KaiA3), Slr1783 (Rre1)/NarL (*E. coli*, UniProtKB - P0AF28), and KaiA was achieved with MEGA X[106,107] using the above constructed alignments. For all alignments, a neighbor-joining tree and maximum likelihood tree were constructed and compared. To construct neighbor-joining trees, 1000 bootstrap iterations with a p-distance substitution model and a gamma distribution with three gamma parameters were used. To construct maximum likelihood trees, an initial tree was constructed using the maximum parsimony algorithm. Further trees were constructed using 1000 bootstrap iterations with an LG-G substitution model, a gamma distribution with three gamma parameters, and nearest-neighbor-interchange (NNI) as the heuristic method.

### Yeast two-hybrid assay

AH109 yeast cells (Clontech) were used for YTH experiments. Transformation of yeast cells was performed according to the manufacturer's guidelines using the Frozen-EZ Yeast Transformation Kit (Zymo Research). Genes of interest were amplified from wild-type genomic DNA using Phusion Polymerase (NEB), according to the manufacturer's guidelines. The indicated restriction sites were introduced using oligonucleotides listed in Table S1A. Vectors and PCR fragments were cut with the respective restriction enzymes, and the gene of interest was ligated into the vector, leading to a fusion protein with a GAL4 activation domain (AD) or GAL4 DNA-binding domain (BD) either at the N- or C-terminus. All constructed plasmids are listed in Table S1B. The detailed protocol for the growth assay can be found at https://doi.org/10.17504/protocols.io.wcnfave[26]. Successfully transformed cells were selected on a complete supplement mixture (CSM) lacking leucine and tryptophan (-Leu -Trp) dropout medium (MP Biochemicals) at 30 °C for 3–4 days. Cells containing bait and prey plasmids were streaked on CSM lacking leucine, tryptophan, and histidine (-Leu -Trp -His) dropout medium (MP Biochemicals) with the addition of 12.5 mM 3-amino-1,2,4-triazole (3-AT, Roth) and incubated for 6 days at 30 °C to screen for interactions.

### Expression and purification of recombinant Kai proteins

*Synechocystis* KaiB3, KaiB1 and *Synechococcus* KaiA (plasmids kindly provided by T. Kondo, Nagoya University, Japan) were produced as GST-fusion proteins in *E. coli* BL21(DE3) as described at https://doi.org/10.17504/protocols.io.48ggztw[26]. Briefly, proteins were purified by affinity chromatography using glutathione-agarose 4 B (Macherey and Nagel), and the N-terminal GST-tag was removed using PreScission Protease (Cytiva) prior to elution of the untagged proteins from the glutathione resin. *Synechocystis* KaiC3 was produced with an N-terminal- Strep-tag (Strep-KaiC3) in *E. coli* Rosetta-gami B (DE3) cells and purified via affinity chromatography using Strep-Tactin XT superflow (IBA-Lifesciences) (see https://doi.org/10.17504/protocols.io.meac3ae[26]). The *Synechocystis* ORF *sll0485*, encoding KaiA3, was inserted into the vector pET22b to create a C-terminal His6-fusion. KaiA3-His6 was expressed in *E. coli* Tuner (DE3) cells and purified by immobilized metal affinity chromatography (IMAC) using PureProteome™ Nickel Magnetic Beads (Millipore). For a detailed protocol, see https://doi.org/10.17504/protocols.io.bu5bny2n. Recombinant proteins were stored at −80 °C in buffer containing 20 mM Tris, pH 8.0, 150 mM NaCl, 0.5 mM EDTA, 5 mM MgCl$_2$, and 1 mM ATP.

### KaiC3 phosphorylation in in vitro assays and liquid chromatography-mass spectrometry (LC-MS/MS)

Recombinant Strep-KaiC3 purified from *E. coli* exists mainly in its phosphorylated form (KaiC3-P). Fully dephosphorylated Strep-KaiC3 (KaiC3-NP) was generated by incubating the protein for 18 h at 30 °C in assay buffer (20 mM Tris, pH 8.0, 150 mM NaCl, 0.5 mM EDTA, 5 mM MgCl$_2$, and 1 mM ATP). The autokinase activity of KaiC3-NP was investigated by incubating 0.2 µg/µl KaiC3 for 16 h at 30 °C in 20 µl assay buffer in the presence or absence of 0.1 µg/µl KaiA3-His6, KaiB3, KaiB1 and *Synechococcus* KaiA, respectively. Aliquots of 10 µl were taken before and after incubation at 30 °C, and the reaction was stopped with SDS sample buffer. Samples were stored at −20 °C prior to application to a high resolution LowC SDS gel (10% T, 0.67% C)[108] using the Hoefer Mighty Small II gel electrophoresis system and Tris-Tricine running buffer (cathode buffer: 100 mM Tris, 100 mM Tricine, 0.1 % SDS, pH 8.25; anode buffer: 100 mM Tris, pH 8.9, according to Schägger and von Jagow[109]). Gels were stained with Coomassie Blue R.

For the 48–60 h assay, pools containing 0.2 µg/µl (3.4 µM) KaiC3-NP, 0.1 µg/µl KaiB3 (7.4 µM) and various concentrations of KaiA3-His6 (corresponding to 0.5–8.4 µM) were prepared in assay buffer supplemented with 5 mM ATP, split in 10 µl aliquots for the desired timepoints and stored at −80 °C. Samples were thawed on ice for 10 min prior to incubation at 25 °C, 30 °C, or 35°C for different time periods. The reaction was stopped at specific time points by adding SDS sample buffer. Samples were stored at −80 °C prior to application to a LowC SDS gel (10% T, 0.67% C)[27] using the Biorad Mini PROTEAN gel electrophoresis system and Tris-glycine running buffer (25 mM Tris, 192 mM glycine, 0.1 % SDS, according to Laemmli[110]). The gels were stained with ROTI®Blue quick stain. In Tris-glycine buffer, three KaiC3 bands were separated, whereas two KaiC3 bands were separated in Tris-tricine buffer. Gels were imaged using a Bio-Rad's ChemiDoc XRS+ Imaging System, and densitometric analysis was performed in ImageLab 6.1. (Bio-Rad). The ratio of PP-KaiC3 to total KaiC3 was calculated in Excel and plotted as the average using GraphPad Prism 10.2.3. To evaluate the band patterns of PP-KaiC3 and NP-KaiC3, KaiC3 was incubated with 10U/µl Lambda phosphatase (NEB) and 1 mM MnCl$_2$ for 14–18 h at 30 °C. As a control, Lambda phosphatase activity was blocked by the addition of PhosSTOP (Roche), or PhosSTOP (Roche) and 10 mM vanadate.

For LC-MS/MS analysis of KaiC3 phosphorylation sites, Strep-KaiC3 and KaiA3 were co-incubated in vitro as described above. Samples were taken directly after mixing and after 2 and 6 h of incubation, and separated by SDS-PAGE. For each sample, gel regions containing proteins of the size of Strep-KaiC were cut using a scalpel. For the 6 h time point, a gel region at the potential size of the Strep-KaiC3/A3 complex was also extracted. In-gel protein digestion with trypsin was performed according to the protocol described by Shevchenko et al.[111]. In brief, gel bands were treated with dithiothreitol and subsequently with iodoacetamide to reduce disulfide bonds and irreversibly alkylate the resulting cysteine thiol groups. Proteomics grade trypsin (Promega) was added to digest proteins overnight at 37 °C. The generated peptides were extracted and purified using the stage tip protocol[112]. Of the resulting peptide solution, 20% was used for nanoLC-MS/MS analysis. Therefore, peptides were separated in a 37 min reverse-phase linear gradient and directly ionized in an online coupled ESI source upon elution for analysis on a Q Exactive HF mass spectrometer (Thermo Fisher Scientific) operated in data-dependent acquisition mode. The 12 most abundant multiply charged ions in each full scan were separately fragmented by HCD, and the generated fragment ions were analyzed in consecutive MS/MS scans. Raw data files were processed using MaxQuant software (version 1.5.2.8) and default settings. Phosphorylation of Ser, Thr, and Tyr was defined as a variable modification. The acquired m/z spectra were searched against the proteome databases of *Synechocystis* and *E. coli* (downloaded from Cyanobase and Uniprot, respectively). Annotated MS/MS spectra were visualized using the MaxQuant integrated viewer.

### Clear native protein PAGE and immunodetection

Kai proteins (containing 0.2 µg/µl dephosphorylated Strep-KaiC3, 0.1 µg/µl KaiA3-His6, KaiB3, KaiB1 or *Synechococcus* KaiA, respectively) were incubated for 16 h at 30 °C in phosphorylation assay buffer, followed by separation of the native proteins in 4–16% native PAGE at 4 °C using a clear native buffer system (Serva) without anionic dye. Thus, only proteins with a pI<7 at physiological pH were separated. Protein bands were visualized with Coomassie staining (ROTIBlue Quick, Carl Roth) or immunodetected with a monoclonal anti-6x-His Tag antibody conjugated to HRP (MA1-21315-HRP, Thermo Fisher, LOT number YH374751, 1:2000 diluted) and imaged using a ChemiDoc XRS+ Imaging System (BioRad). A detailed protocol can be found at https://doi.org/10.17504/protocols.io.bu67nzhn.

### Strains and growth conditions

Three laboratory strains of *Synechocystis* PCC 6803 were used in this study: PCC-M (resequenced[29]), 'Uppsala' (kindly provided by Pia Lindberg, Uppsala University), and 'Chicago' (kindly provided by Dr. Amin Omairi-Nasser, University of Chicago).

Wild-type *Synechocystis* (PCC-M), the derived deletion strains Δ*rpaA*[38], Δ*kaiC3*[26], Δ*kaiA3*, and Δ*kaiA3B3C3* (Fig. S1), and complementation strain Δ*kaiA3/kaiA3* (Fig. S1) were cultured photoautotrophically in BG11 medium[113] supplemented with 10 mM TES buffer (pH 8) under constant illumination with 75 µmol photons m$^{-2}$ s$^{-1}$ white light (Philips TLD Super 80/840) at 30 °C. The cells were grown in Erlenmeyer flasks with constant shaking (140 rpm) or on plates (0.75% Bacto-Agar; Difco) supplemented with 0.3% thiosulfate. For photomixotrophic experiments, 0.2% glucose was added to the plates. For chemoheterotrophic growth experiments in complete darkness, *Synechocystis* cells were spotted at different dilutions on BG11 agar plates containing 0.2% glucose and incubated either mixotrophically for three days with continuous illumination or chemoheterotrophically in the dark for 26 days.

Wild-type *Synechocystis* (Chicago) and its derived *kaiC* mutant strains were grown in BG-11 M liquid medium supplemented with 20 mM HEPES (pH 8.0) at 30 °C with shaking (165 rpm) in a Percival incubator under a light intensity of ~50 µmol m$^{-2}$s$^{-1}$, provided by cool white fluorescent light bulbs (Philips Alto II, USA). The *kaiC* mutant strains were grown in the presence of the appropriate antibiotics.

Wild-type *Synechocystis* (Uppsala) and it's derived deletions strains Δ*kaiA1B1C1*[53], Δ*kaiA3B3C3*, Δ*kaiA3*, Δ*kaiB3*, Δ*kaiC3*, Δ*kaiA3B3*, Δ*kaiA3C3*, Δ*kaiB3C3* were grown in BG11 medium supplemented with 10 mM TES buffer (pH 8) in a Multitron Infors HT® Incubator at 30 °C, 0.5% CO$_2$, and 75% humidity under constant illumination with 80 µmol photons m$^{-2}$ s$^{-1}$ of white light. Growth occurred on plates (0.75% Bacto-Agar; Difco) or in Erlenmeyer flasks with shaking at 150 rpm. For backscatter measurements, cultures were incubated in a biolector equipped with a light array module (Beckman Coulter) (see *Detection of* in vivo *oscillations* via *backscatter analysis*).

### Construction of mutants of the KaiC3 based clock system

**Mutants constructed in the *Synechocystis* (PCC-M) background strain.** To construct the *kaiA3* (*sll0485*) deletion strain, *Synechocystis* wild-type cells were transformed with the plasmid pUC19-Δ*sll0485*. For plasmid construction, PCR products were generated using the oligonucleotides P13/P14 and pUC19 as template, P15/16 and P19/20 with genomic *Synechocystis* wild-type DNA as template and P17/25 with pUC4K as template. Homologous recombination led to the replacement of the *kaiA3* gene with a kanamycin resistance cassette (Fig. S1). For genomic complementation of the Δ*kaiA3* strain, cells were transformed with the plasmid pUC19-Δ*sll0485*-compl. Overlapping fragments were generated using oligonucleotides P15/28 and P24/32 with genomic *Synechocystis* wild-type DNA as template, P13/26 and pUC19 as template, and P22/23 and the vector pACYC184 as template. In the resulting complementation strain Δ*kaiA3/kaiA3*, the kanamycin

resistance cassette was replaced with *kaiA3*, and a chloramphenicol resistance cassette was introduced downstream of the *kaiB3* gene (Fig. S1). For the triple-knockout mutant Δ*kaiA3B3C3*, Δ*kaiC3* cells were used as the background strain for transformation with the pUC19-Δ*kaiA3B3* plasmid. PCR products were generated using the oligonucleotides P13/26 and pUC19 as template, P17/27 and pUC4K as template, P15/16 and P25/28 with genomic *Synechocystis* wild-type DNA as template. The operon *kaiA3kaiB3* was replaced with a kanamycin resistance cassette (Fig. S1). Complete segregation of the mutant alleles was confirmed using PCR. For the Δ*kaiA3* strain, oligonucleotides P15/29 were used. Segregation of the complementation strain was confirmed by PCR with P15/29, P30/31, and P19/32. For the triple knockout mutant Δ*kaiA3B3C3*, deletion of the *kaiA3B3* operon was confirmed by PCR using the primer pairs P15/33 and P19/30. The *kaiA3B3* chromosomal region of the mutants is shown in Fig. S1. Ectopic expression of *kaiA3* was achieved in wild-type and Δ*kaiA3* cells after transformation with the plasmid pUR-NFLAG-*sll0485*. The plasmid was constructed via restriction digestion of the vector pUR-N-Flag-xyz, and the PCR product was amplified with the oligonucleotide pair P29-P34 using genomic *Synechocystis* wild-type DNA as a template. Restriction digestion using EcoRI and BamHI was followed by ligation. Successful transformation was confirmed by PCR using P35/36. The oligonucleotides and plasmids used are listed in Table S1.

**Mutants constructed in the *Synechocystis* (Chicago) background strain.** The *kaiC1* gene was amplified from *Synechocystis* genomic DNA using oligonucleotide primers P41/42. The *kaiC1* DNA fragment (1566 bp) amplified by P41/42 was cloned into the pGEMT-Easy plasmid (pGEMT-Easy+ kaiC1 plasmid). A kanamycin gene cassette was amplified using primers P43/44) and PsasA-Nina plasmid (kindly provided by Carl H. Johnson) as a template. The *kaiC1* ko plasmid was constructed by Gibson assembly using the following DNA fragments: a KpnI-digested pGEMT-Easy+ kaiC1 plasmid backbone and a 1340 bp sequence fragment of the kanamycin cassette, resulting in *pGEMT-ΔkaiC1*. The pUC19_Δ*kaiC3* plasmid (table S1) was modified and the chloramphenicol cassette was removed and replaced with a kanamycin cassette. The knockout plasmids were incorporated into the cyanobacterial genome by natural transformation[30]. Fully segregated mutants were generated by streaking the transformants multiple times with appropriate antibiotics.

**Mutants constructed in the *Synechocystis* (Uppsala) background strain.** To generate the plasmid pUC19-Δ*kaiB3*, PCR products containing overlapping fragments were produced using P37/38 and P25/28 and wild-type DNA as a template. The kanamycin resistance cassette, substituting *kaiB3*, was amplified from pUC4K using oligonucleotides P27/39. The pUC19 backbone was amplified using P26/40. The plasmid was assembled by ligating overlapping fragments. Wild-type *Synechocystis* (Uppsala) was transformed with plasmids pUC19_Δ*sll0485*, pUC19_Δ*kaiB3*, pUC19_Δ*kaiC3*, or pUC19_Δ*kaiA3B3* to generate deletion strains Δ*kaiA3*, Δ*kaiB3*, Δ*kaiC3*, Δ*kaiA3B3*, respectively (see table S1). Transformation of *Synechocystis* Δ*kaiC3* with plasmids pUC19_Δ*kaiA3B3*, pUC19_Δ*sll0485* or pUC19_Δ*kaiB3* was performed to obtain Δ*kaiA3B3C3*, Δ*kaiA3C3*, and Δ*kaiB3C3*, respectively. Segregation was achieved by selection on plates with appropriate antibiotics.

**Detection of in vivo oscillations via SDS-PAGE and immunodetection**
Wild-type *Synechocystis* sp. PCC 6803 strain (Chicago)and derived *kaiC* mutant strains were exposed to two LD cycles (12 L:12D) to synchronize the circadian clock, then released to constant light conditions (LL) (~50 μmol m$^{-2}$ s$^{-1}$) and cultures were harvested (centrifuged at 2800 × *g* for 10 min at room temperature) every 4 h. Cell pellets were flash frozen in liquid nitrogen and stored at −80 °C until samples were

processed for western blot analysis. Total protein extraction and western blot analysis were performed as previously described[23]. In brief, each frozen cell pellet was resuspended in 200 μl of lysis buffer (8 M urea, 20 mM HEPES pH 8.0) and cells were broken by vigorous vortexing (30 s vortex and 1 min cooling in ice for 7 cycles) with glass beads (0.1 mm, acid-washed). The supernatant fraction containing total protein was collected after centrifugation (1000 × *g* for 3 min at 4 °C) the cell suspension. The protein concentration was measured using the Bradford assay. Equal amounts of total protein (8 μg) were mixed with 4X DTT-containing SDS polyacrylamide gel electrophoresis (SDS-PAGE) sample buffer. The samples were heated at 95 °C for 4 min and were loaded onto big-format SDS-PAGE gels (10%). The gel was run for 4.5 h at a constant current of 35 mA per gel and 12 °C for protein separation. The protein samples were transferred onto PVDF membranes, blocked with 4% w/v nonfat dry milk/ Tris-buffered saline with 0.1% Tween-20 (TBS-T) for 2 h at room temperature. Membranes were then incubated overnight at 4 °C with anti-KaiC1 (1:3750 dilutions in TBST) and anti-KaiC3 antibody (1:7500 dilutions in TBST). Membranes were washed 3 times (20 minutes each at room temperature) in TBST and finally probed with Goat anti-rabbit IgG (H + L) Secondary Antibody, HRP (Thermo Fisher Scientific, 1:10000 dilution). Chemiluminescent detection was performed using Pierce SuperSignal West Pico detection reagent (Thermo Scientific). Blots were photographed using a ChemiDoc MP Imaging System (Bio-Rad). Total protein abundance and the ratio of PP-KaiC to total KaiC protein were estimated by densitometry of the blot images and plotted using GraphPad Prism 9.5.1.

**Detection of in vivo oscillations via Phos-tag SDS-PAGE and immunodetection**
To analyze the in vivo phosphorylation of KaiC3, *Synechocystis* (PCC-M) wild type, Δ*kaiA3*, and Δ*kaiC3* cells were cultivated in BG11 or copper-depleted medium for *kaiA3* overexpression. The cells were grown in two consecutive 12 h LD cycles followed by 24 h of LL conditions. After the initial 12 h LD cycle, 10 ml of cells were collected every 6 h for analysis. The cells were cooled in liquid nitrogen for 5 s and harvested by centrifugation (3220 × g, 2 min, 4 °C). The pellet was frozen in liquid nitrogen and stored at −20 °C until further processing. To lyse the cells, the pellets were resuspended to an OD$_{750}$ of 25 in phosphorylation buffer (50 mM NaOH-HEPES pH 7.5, 300 mM NaCl, 0.5 mM Tris-(2-carboxyethyl)-phosphine, 10 mM MgCl$_2$). The cells were disrupted twice in a cell mill at 30 Hz for 1 min at 4 °C, using glass beads. The crude cell extract was obtained by centrifugation (500 × g, 1 min, 4 °C). For mobility shift detection of phosphorylated and dephosphorylated KaiC3, a Zn$^{2+}$-Phos-tag® SDS-PAGE assay (Wako Chemicals) was used. A 9% SDS-PAGE gel containing 25 μM Phos-tag acrylamide was prepared and 12 μL of the cell extract was run at 150 V for 3 h at 4 °C. Proteins were blotted onto a nitrocellulose membrane (Amersham Protran®) using wet blotting. The membranes were blocked with 5% dry milk in TBS-T for 1 h at room temperature. Immunodetection was performed using the αKaiC3[28] antibody(1:7500 in TBS-T) at 4 °C overnight, and subsequently with anti-rabbit αHRP (Thermo Fisher Scientific Inc., USA) antibody (1:20000 in TBS-T) for 2 h at room temperature. Between the steps, the membrane was washed with TBS-T for 10 min at room temperature. Protein detection was performed using Pierce ECL western blotting Substrate (Thermo Scientific). The blots were visualized using a Fusion SL chemiluminescence detector (Vilber Lourmat). The ratios of phosphorylated to non-phosphorylated KaiC3 were quantified via mean gray value measurements (ImageJ[114]) and plotted as average with GraphPad Prism 10.2.3.

**Detection of in vivo oscillations via backscatter analysis**
Prior to monitoring the backscatter properties of wild-type *Synechocystis* (Uppsala variant) and the indicated *kai* deletion mutants, cells were grown for 10 days under LL as two subsequent pre-cultures in

BG11 medium without antibiotics. For pre-culture 1, cells from agar plates were inoculated in 20 ml medium and grown in a 100 ml Erlenmeyer flask under constant illumination with 80 μmol photons $m^{-2} s^{-1}$ in a Multitron Infors HT® incubator set to 30 °C, 75% humidity, 150 rpm and 0.5% $CO_2$ supply. After 7 days, cultures reached an $OD_{750nm}$ of ~4-6. For pre-culture 2, the cells were diluted to $OD_{750}$ nm of approx. 0.6 (Fig. 5a–f) or 0.7 (Fig. S10) in a total volume of 100 ml medium and incubated for 3 further days in 250 ml Erlenmeyer flasks under the same conditions. The cells reached an $OD_{750nm}$ of ~1.8-2.4. At the start of the experiment, cells from pre-culture 2 were diluted to an $OD_{750nm}$ of 0.9 with BG11 without antibiotics. Per strain, 4-5 wells of a 48-well FlowerPlate MTP (Beckman Coulter) were filled with 1 ml cells each (originating from the same pre-culture), the plate was sealed with gas-permeable foils and transferred to a BioLector XT microbioreactor equipped with a Light Array Module (LAM) (Beckmann Coulter). This process took approx. 1-2 hours. Cells were incubated at 30 °C and 600 rpm, with a flow of 20 ml $min^{-1}$ of humidity-controlled air with 1% $CO_2$. The LAM module was programmed to roughly mimic the light spectrum of LEDs in the Infors HT® incubator and was set to a constant illumination of ~80 μmol $m^{-2} s^{-1}$. Backscatter at 730 nm (gain 6) was measured every 5 min and analyzed for 84 h. The BioLection and LUA protocols can be found together with the raw data as Supplementary Data S4. All optical densities mentioned above were measured in a Specord 200 plus (Analytic Jena) using dilutions that allowed measurements in the range of $OD_{750nm}$ 0.2-0.7.

Processing of the data has been done using Python (version 3.9.16), with packages: pandas (version 1.5.3), matplotlib (version 3.7.1), regex (version 2022.7.9), numpy (version 1.24.3), scipy (version 1.10.1), scikit-learn (version 1.2.2). The script[53] was modified and is available at https://github.com/flo-sti/cyano-backscatter[115]. The backscatter data were imported from.xlsx files and raw data were plotted. To isolate the oscillation signal, we removed growth effects using sklearn.LinearRegression. A fourth degree polynomial was fitted to the observed raw backscatter values, excluding data from the first 3 h. Regression values were predicted using sklearn.predict, and subtracted from the observed raw backscatter values. The resulting signal was normalized by subtracting the arithmetic mean and smoothed with the numpy.convolve function with a kernel size of 40 and the mode 'same'. The smoothened signal was used for further graphical and statistical analysis.

To determine the period of the oscillation for each strain, we first attempted to fit an equation for a simple harmonic oscillation of the following form to the data (A: amplitude, ω: angular frequency, φ:phase angle):

$$y(t) = A\cos(\omega t + \varphi) \qquad (1)$$

The fit was performed with the scipy.optimize.fit_curve function in Python, and the resulting parameters were used to generate a curve that could be plotted along with the signal. The boundaries for the fit_curve function were chosen as follows: amplitude: $0 \leq A \leq max(data) - mean(data)$; angular frequency: $2*\pi/(maximum\ estimated\ period) \leq \omega \leq 2*\pi/(minimum\ estimated\ period)$ with the minimum estimated period between 18 h and 35 h and the maximum estimated period between 26 h and 45 h depending on the strain; phase angle: $2*\pi/12 < = \varphi < = 2*\pi/1$. The formula and amplitude estimates were based on the study by Santos et al.[116]. This fit performed well for the wild type, Δ$kaiA1B1C1$, and Δ$kaiB3$ mutants and was used to derive the period of these strains, but the fit failed to accurately capture the phenotype of the $kaiA3$ and $kaiC3$ knockouts.

To calculate the length of the first period for strains that could not be fitted with a simple harmonic oscillation, the scipy.find_peaks function (distance=150; Height=50) was used to identify the first trough and peak. Since the time difference between these values is only half of a period, the result was multiplied by two and the absolute

value was taken. This method was chosen to measure the period length because it was sometimes impossible to determine the later peaks using the same parameters for each strain. However, the first trough and peak were detected reliably. The values of the length of the first period of all strains for all three experiments were collected, and the arithmetic mean, median, and standard deviation were calculated. Pairwise statistical tests were performed for each strain combination. First, a Levene's test was performed to determine whether the compared strains had the same variance. If the variance was the same, a student's t-test was performed. If the variances differed significantly (p < 0.05), a Welch's t-test was performed. Differences between two strains were considered significant if the p-value of the statistical test was <0.05.

To determine the phase shift and relative amplitude after synchronization in comparison to the wild type for all strains, independent of whether they displayed an undampened or dampened oscillation, peaks were identified using the scipy.find_peaks function with a distance of 150, width of 65, and a minimum height of the mean of the signal + 20 % of the maximum value. The time point and height of the first peak were determined, and for each strain, the arithmetic mean and standard deviation (SD) of the 4-5 replicates within one experiment were calculated. Peak detection in the normalized backscatter data of the $kaiA1B1C1$ mutant was difficult due to the low amplitude. In the first two experiments, we observed a broad peak with a slightly higher bump in the front compared with the harmonic fit (Fig. 5e) and used the first detectable peak to calculate the phase shift and amplitude. For each experiment, the results were correlated with the wild type as follows:

$$phase\ shift = peak\ time[mutant] - peak\ time[wild\ type]\ with \qquad (2)$$

$$error\ boundaries = SD[mutant] + SD[wild\ type] \qquad (3)$$

$$relative\ amplitude = \frac{peak\ hight[mutant]}{peak\ hight[wild\ type]}\ with \qquad (4)$$

$$error\ boundaries = \frac{SD[mutant]}{peak\ hight[mutant]} + \frac{SD[wild\ type]}{peak\ hight[wild\ type]} \qquad (5)$$

## Screening of KaiC3 and KaiC1 binding partners by immunoprecipitation-coupled liquid chromatography mass spectrometry (LC-MS/MS)

*Synechocystis* (PCC-M) WT/FLAG-*kaiC3*, WT/FLAG-*kaiC1*, and WT/FLAG-*sfGFP* (control) strains were cultivated in BG11 medium (100 ml, copper depleted), harvested by centrifugation at 6000 × *g* for 10 min at 4 °C and resuspended in purification buffer (50 mM HEPES/NaOH (pH 7.5), 5 mM $MgCl_2$, 25 mM $CaCl_2$, 10 % (v/v) glycerol, 150 mM NaCl, 5 mg $ml^{-1}$ 6-aminohexaonic acid, 1 mM 4-(2-aminoethyl)benzenesulfonyl fluoride hydrochloride, 4 mM *p*-aminobenzamidine, 1 mM ATP). Cells were disrupted in a mixer mill, followed by solubilization with *n*-dodecyl-β-maltoside (detergent-to chlorophyll ratio 20:1) for 1 h at 4 °C. The supernatant was used for FLAG purification in pull-down assays with Anti-Flag® M2 Magnetic Beads (Sigma-Aldrich), following the manufacturer's protocol. The resulting elution fractions were loaded onto a NuPAGE™ Bis-Tris Gel and run following the manufacturer's protocol (Invitrogen). Protein bands were allowed to migrate for only a short distance of approximately 10 mm. After staining the gel for 60 min with InstantBlue™ (Expedeon), the protein-containing gel regions were excised. Two independent replicates were produced for each condition (KaiC3, KaiC1, and control pull-down). In-gel protein digestion with trypsin was performed as described above, and the resulting peptide solutions were purified using stage tips. Approximately 20% of the sample was subjected to nanoLC-MS/MS analysis as

described above on a Q Exactive HF mass spectrometer (Thermo Fisher Scientific) operated in data-dependent acquisition mode. Raw data of KaiC3 or KaiC1 pull-downs were separately processed using the MaxQuant software (version 1.5.2.8) embedded MaxLFQ algorithm as described by Cox et al.[117]. Raw spectra were searched against the proteome databases of *Synechocystis* and *E. coli* (downloaded from Cyanobase and Uniprot, respectively) and bait protein sequences. Significantly enriched proteins were identified using Perseus software (version 1.6.5.0) significance B analysis with a *p*-value of 0.01.

### Reporting summary

Further information on research design is available in the Nature Portfolio Reporting Summary linked to this article.

## Data availability

Raw and processed data generated in this study are available on figshare: Raw data: https://doi.org/10.6084/m9.figshare.25218143. Processed Data: https://doi.org/10.6084/m9.figshare.25218137, Alignments: https://doi.org/10.6084/m9.figshare.25218122, Phylogeny: https://doi.org/10.6084/m9.figshare.25218134), processed KaiA3 hits are also available as Supplementary Data S1. The mass spectrometry proteomics data generated in this study have been deposited in the ProteomeXchange Consortium database (http://proteomecentral.proteomexchange.org) via the PRIDE partner repository[118], with the dataset identifier PXD042846 (analysis of KaiC3 phosphorylation) https://ftp.pride.ebi.ac.uk/pride/data/archive/2024/07/PXD042846, PXD042845 (screening of KaiC3 and KaiC1 binding partners) https://ftp.pride.ebi.ac.uk/pride/data/archive/2024/07/PXD042845, and summarized data are available as Supplementary Data S2 and S3. Datasets S1 to S4 are available as Supplementary Data. Source data are provided with this paper.

## Code availability

The script for reciprocal BLAST analysis is available at https://github.com/schmelling/reciprocal_BLAST/blob/master/notebooks/. The script for backscatter data analysis is available at https://github.com/flo-sti/cyano-backscatter[115]. Raw data, Biolection and LUA protocols are available as Supplementary Data S4.

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

## Acknowledgements

The authors thank all members of the Research Unit FOR2816 'SCyCode' (The Autotrophy-Heterotrophy Switch in Cyanobacteria: Coherent Decision-Making at Multiple Regulatory Layers) for fruitful discussions and the German Research Foundation (DFG) for financial support (project number 397695561, A.W., B.M.). This work was further supported by grants from the DFG (WI2014/5-3 (A.W.); WI2014/10-1 and 10-2 (A.W.); AX 84/1-3 (I.M.A.); MA 4918/4-1 (B.M.)), and SFB1535 - Project ID 458090666 (I.M.A.). I.M.A. and A.Wie were further supported by the DFG under Germany's Excellence Strategy – EXC-2048/1, project ID 390686111 (CEPLAS). NIH R01 GM135382 provided funding to M.J.R. We thank Pauline Morys, Katerine Cheronis, Annika Klopp, Isabell Bleile, and Werner Bigott for technical assistance. Figure 8 was created using BioRender.com.

## Author contributions

C.K., I.M.A., A. Wie, and A.W. designed the study. C.K., N.M.S., A.P., P.S., N.M.Sche., K.N.S., A. Wie, G.K.P., F.P.S., P.K., and L.B. performed and analyzed the experiments. All authors interpreted and discussed the data. C.K., N.M.S., A.P., P.S., G.K.P., M.J.R., B.M., A. Wie, I. M.A., and A.W. wrote the manuscript.

## Funding

## Competing interests

The authors declare no competing interest.
