## [Peer Review File · Nature Communications]

Two KaiABC systems control circadian oscillations in one cyanobacteriumReviewer #1 (Remarks to the Author):

The manuscript by Kobler et al. provides insight into the presence of two circadian oscillators in *Synechocystis*. Authors discovered an ortholog of KaiA1 protein by using different bioinformatic tools and named the protein KaiA3. Using yeast two hybrid experiment and native polyacrylamide gel analysis, authors verified the interaction of KaiA3 and KaiC3. Recombinant KaiA3 phosphorylated KaiC3 and the phosphorylation of KaiC3 was abolished when KaiB3 was added in the reaction mixture. Taking all the three purified proteins (KaiA3, B3 and C3), authors reconstituted the circadian clock reaction in vitro and claimed that the phosphorylation of KaiC3 oscillated in a circadian manner (i.e., with 24 hr period). Authors further checked the oscillation of KaiC3 phosphorylation in *Synechocystis* cells by western blot using KaiC3 specific antibody. In both in vitro and in vivo, authors showed the phosphorylation of KaiC3 oscillated in a circadian manner. So, in addition to the canonical KaiA1B1C1 (orthologs of *S. elongatus* circadian clock KaiABC proteins), authors claimed a second circadian oscillator (and named it as KaiA3B3C3) does exist in *Synechocystis*. To show the function of the second circadian oscillator, authors did the viability assay in different growth conditions and showed that the kaiA3 null cells were less viable in photomixotrophic condition (continuous light, +glucose) and heterotrophic condition (dark+glucose).

It is interesting that Authors discovered a kaiA like protein named KaiA3 that could phosphorylate KaiC3 (the ortholog of KaiC1). However, it is questionable whether the experiments provided pushes the current manuscript for acceptance in Nature communication. No data were given to support the existence of KaiA1B1C1 oscillator. The core figure of the manuscript (Fig 3A and 3C) is very confusing and should be substantially improved by the authors to support their claim for the existence of the second circadian oscillator, i.e., KaiA3B3C3. It has been known that the KaiC3 (which is the part of the second oscillator) null cells have viability issues in continuous dark. Though in this manuscript, some preliminary growth results were given for KaiA3 null cells in different growth conditions it is not clear why the cell viability are different in KaiA3 vs KaiC3 vs KaiA3B3C3 in some condition(s) even if they jointly drive the second oscillator! How the A3B3C3 system regulates the physiology?

Here are my detailed comments:

The focus of this manuscript is to show that there are “two circadian oscillator” exist in *Synechosystis*: one circadian oscillator is driven by Kai A1B1C1 and the second oscillator is driven by Kai A3B3C3. Authors, only tried to show the oscillation in KaiC3 phosphorylation, but have not provided any data for KaiC1 phosphorylation oscillation either in vitro or in vivo. Are those data published anywhere that I do not know?

Authors cited the recent work from Johnson lab (Zhao et al, 2023) where a reporter strain was made to show the bioluminescence oscillation. The rhythm in the oscillation in that reporter strain was gone in kaiC1 null as well as in kaiC3 null cells. The circadian clock study in *Synechocystis* varies a lot from lab to lab. For example, Kucho et al., 2005 showed circadian clock exists in *Synechocystis*, however Beck et al, 2014 claimed the absence of circadian rhythm in *Synechocystis*. Similarly, work from the authors' lab could not generate kaiB2C2 null but Johnson lab could make kaiB2C2 null cells. Indeed lot of data are dependent on the background strain. That's why it is advisable to check whether the KaiA1B1C1 cluster can generate an oscillation in KaiC1 phosphorylation in the background strain (KaiC1 western blot) that the

authors used for this study and also in vitro. Are those the two oscillators in phase or oscillate with different amplitude and/or period?

Figure 3A and 3C are confusing and can be improved a lot. Please clarify.

In fig 3A, authors used purified recombinant proteins (KaiA3, B3 and C3) and checked the circadian oscillation of phosphorylated KaiC3 (PPKaiC3) for 48 hours and claimed KaiC3 phosphorylation showed sustained oscillation with a 24 hrs rhythm (when KaiA3 is either 1.4 μ M or 2.8 μ M) that is a characteristic feature of a circadian oscillator.

Looks like in Fig 3A, KaiC3 phosphorylation peaked at 6th hour and dephosphorylated (trough) at 18th hour. At 24th hour, the PPKaiC3 is higher than the 18th hour. This suggests the phosphorylation cycle is complete at 18th hour. However, if you look at the phosphorylation peaks at 6th hour and 30th hour, it does look like a 24 hour cycle. But, in that case it would be clear if the experiment was extended up to 54th hour to see the second peak at 54th hour to confirm a 24 hour cycle.

Is the oscillation in PP-KaiC is dying in the second cycle (please look at 18th to 36th hour compared to 0-18th hour)? Is it not a sustained oscillator? Authors have used three replicates (marked by dots) for each time point, if you take the average of all the dots at each time point and plot, does the trace look like an oscillation (from 18th hour onward)? The reason I am bringing this as the amplitude of PPKaiC3 oscillation in the second cycle is very low and the replicates marked by three dots at each time points have a wide range of variation; check the dots at 24th and 30th hour time points. A true circadian clock oscillator is temperature compensated. KaiC3 ATPase activity is not temperature compensated. Will KaiC3 mediated oscillation be considered as circadian? Any comments on that?

Figure 3C: Using KaiC3 specific antibody, authors checked the oscillation of P-KaiC3 in *Synechocystis* cells. They mentioned (line 354, page 11) the two prominent bands which are absent in Δ KaiC3 strain most probably reflect two phosphorylation states of KaiC3. They also marked the very upper band (red arrow) as the fully phosphorylated one and one of the lower bands (blue arrow) is the unphosphorylated form. I am confused with the upper band that is marked with PP-KaiC3. The topmost band (marked for PPKaiC3) should not show up in *kaiA3* null cells? Am I wrong? How come KaiC3 gets phosphorylated in absence of KaiA3? If that band is showing up in *kaiA3* null, then that does not represent the true PP-KaiC3 band. So, how that band was used for quantification to check the oscillation in KaiC3 phosphorylation? Again, authors took the ratio of fully phosphorylated KaiC3 (PP-KaiC3) to non-phosphorylated KaiC3 (N-KaiC3) for quantification. But, why the authors only took the PP-KaiC3 band but did not consider both the phosphorylated bands for quantification?

Reviewer #2 (Remarks to the Author):

The manuscript entitled "Two circadian oscillators in one cyanobacterium" by Köbler et al, is a worthy attempt to tackle a really interesting question, namely whether two independent

circadian oscillators can exist and operate separately in a single cell; in this case, a bacterium. To the knowledge of this Reviewer, this question has only been addressed persuasively once before, namely in experiments on the classic dinoflagellate system where two separate circadian rhythms could be monitored under free-running constant conditions and the researchers found a significant difference in the free-running periods of the two rhythms as their underlying oscillators ran independently (Nature 362:362-364, 1993; which is a reference that the authors inexplicably do not cite).

Unfortunately, the authors of the current manuscript seem to not appreciate how difficult it is to persuasively make the case for two independent oscillators in a single cell. What they show here is that the KaiA3B3C3 proteins are capable of promoting a weak oscillation of KaiC3 phosphorylation in vitro ("weak" because the reported data show a high variability of data points). They then make two key (and unsupported) assumptions:

1. based on the data from the PCC7942 system, the KaiA1B1C1 system can also oscillate by itself in PCC6803 (what's the evidence for this in PCC6803?).

2. that the KaiA1B1C1 system does not interact with the KaiA3B3C3 system in vivo to form a composite single oscillator in vivo. I notice from their 2013 paper that (Fig. 4 of Wiegard et al., 2013) from FLAG-tag pulldowns that the systems do not appear to interact in vivo by, e.g., KaiC monomer exchange. This is a good start, but it doesn't eliminate the possibility of interaction between the two systems in vivo by a different mechanism. Moreover, are they sure that their FLAG-tags have not interfered with functionality of the Kai proteins and/or sterically inhibited monomer exchange? Tags can often inhibit function.

In order for the authors to persuasively claim that there are two circadian oscillators in PCC6803, they need to show either (or both) of the following:

1. In vitro test: do experiments like those shown in Fig. 3C but where KaiA1B1C1 and KaiA3B3C3 are both mixed in the test tube at the same time and the phosphorylation cycles of KaiC1 and KaiC3 are independently monitored (perhaps with antibodies specific for KaiC1 vs. KaiC3) for enough cycles to show that the free-running periods of the KaiA1B1C1 and KaiA3B3C3 systems are independent. This experiment needs to be done at different temperatures to confirm temperature compensation of both systems; temperature compensation must be demonstrated for both systems----otherwise, they are not circadian oscillators.

2. In vivo test: using a luminescence reporter system with reporter luciferases driven by the two systems, (1) KaiA1B1C1 (using promoters from one of these genes fused to luxAB) and (2) KaiA3B3C3 (using promoters from one of these genes fused to luxAB), the authors might be able to show that these two rhythms run with different periods (by analogy to the aforementioned paper in Nature 362:362-364, 1993). This experiment could be most persuasively accomplished using two-color luciferases (e.g., Plant Cell Physiol. 45:109-13, 2004) where one color tracks the KaiA1B1C1 system while the other color tracks the KaiA3B3C3 system simultaneously.

We highly appreciate the very helpful comments on our manuscript and wish to thank the editor and reviewers for carefully reading our manuscript. As outlined below, we have included new data and reorganized the article to improve clarity. Please find the comments from the reviewers followed by our answers in bold.

REVIEWER COMMENTS

##

Reviewer #1 (Remarks to the Author):

The manuscript by Kobler et al. provides insight into the presence of two circadian oscillators in *Synechocystis*. Authors discovered an ortholog of KaiA1 protein by using different bioinformatic tools and named the protein KaiA3. Using yeast two hybrid experiment and native polyacrylamide gel analysis, authors verified the interaction of KaiA3 and KaiC3. Recombinant KaiA3 phosphorylated KaiC3 and the phosphorylation of KaiC3 was abolished when KaiB3 was added in the reaction mixture. Taking all the three purified proteins (KaiA3, B3 and C3), authors reconstituted the circadian clock reaction *in vitro* and claimed that the phosphorylation of KaiC3 oscillated in a circadian manner (i.e., with 24 hr period). Authors further checked the oscillation of KaiC3 phosphorylation in *Synechocystis* cells by western blot using KaiC3 specific antibody. In both *in vitro* and *in vivo*, authors showed the phosphorylation of KaiC3 oscillated in a circadian manner. So, in addition to the canonical KaiA1B1C1 (orthologs of *S. elongatus* circadian clock KaiABC proteins), authors claimed a second circadian oscillator (and named it as KaiA3B3C3) does exist in *Synechocystis*. To show the function of the second circadian oscillator, authors did the viability assay in different growth conditions and showed that the *kaiA3* null cells were less viable in photomixotrophic condition (continuous light, +glucose) and heterotrophic condition (dark+glucose).

It is interesting that Authors discovered a *kaiA* like protein named KaiA3 that could phosphorylate KaiC3 (the ortholog of KaiC1). However, it is questionable whether the experiments provided pushes the current manuscript for acceptance in Nature communication. No data were given to support the existence of KaiA1B1C1 oscillator. The core figure of the manuscript (Fig 3A and 3C) is very confusing and should be substantially improved by the authors to support their claim for the existence of the second circadian oscillator, i.e., KaiA3B3C3. It has been known that the KaiC3 (which is the part of the second oscillator) null cells have viability issues in continuous dark.

Answer: We thank the reviewer for critically reading our manuscript. We are truly grateful for the helpful suggestions on how to improve the manuscript. Based on these comments, we have included new data demonstrating the presence and function of the KaiA1B1C1 oscillator *in vivo* (Fig. 4). To strengthen the claim that KaiA3B3C3 can form an oscillator, we have now included *in vitro* data showing the temperature compensation of the system (Fig. 3d) and extended *in vivo* data showing three free-running cycles of KaiC3 phosphorylation (Fig. 4). These data support our conclusion that there are two oscillators in *Synechocystis*. Based on the new backscatter analysis, we demonstrated that both systems, KaiA1B1C1 and KaiA3B3C3, are required to drive self-sustained 24h rhythms which also helps to better interpret the previous data (Fig. 5). We believe that these two oscillators are directly or indirectly connected. To emphasize this interconnection, we have changed the title of the manuscript accordingly.

Below we address the comments point by point

Though in this manuscript, some preliminary growth results were given for KaiA3 null cells in different growth conditions it is not clear why the cell viability are different in KaiA3 vs KaiC3 vs KaiA3B3C3 in some condition(s) even if they jointly drive the second oscillator! How the A3B3C3 system regulates the physiology?

Answer: Based on the reviewer's suggestion, we have included new data on free-running in vivo oscillations of KaiC1 and KaiC3 phosphorylation in the revised manuscript (see below). Your suggestion to include a focus on the KaiA1B1C1 system (next to KaiA3B3C3), provided valuable input to interpret also the phenotype of the presented mutants in the context of both systems and KaiC3 phosphorylation. By comparing the phenotypes of $\Delta kaiA1B1C1$ ^{1, 2} and $\Delta kaiC3$ mutants¹, we showed that in any mutant in which the *kaiC3* gene was deleted, growth was less affected than after deletion of the *kaiA1B1C1* operon. Manipulation of the KaiA3 level, however, was as detrimental as the knockout of the *kaiA1B1C1* operon, regardless of whether *kaiA3* was deleted or overexpressed. Based on these data, we speculate that the phosphorylation of KaiC3 plays a critical role in phenotypic expression. Furthermore, KaiA3 may have important interactors outside of KaiB3 and KaiC3 that contribute to the phenotype (now summarized in the model in Fig. 8). Beyond this, we agree with the reviewer that we were unable to provide more details on how the KaiC3-based system controls the physiology of *Synechocystis* cells. In Köbler et al.³ and Scheurer et al.⁴, we mainly analyzed an *rpaA* mutant strain in which the KaiC1-based system controlled many aspects of the switch between autotrophic and heterotrophic metabolism. Based on the growth phenotype of *kaiC3* and *kaiA3* strains, which is similar to that of the $\Delta kaiA1B1C1$ strain, we can hypothesize that again the dark metabolism is affected in these mutants. However, as we do not know the output components of the KaiA3B3C3 system, we can only speculate on how the system regulates physiology (see also comments below). Such an analysis is beyond the scope of this study.

Here are my detailed comments:

The focus of this manuscript is to show that there are “two circadian oscillator” exist in *Synechocystis*: one circadian oscillator is driven by Kai A1B1C1 and the second oscillator is driven by Kai A3B3C3. Authors, only tried to show the oscillation in KaiC3 phosphorylation, but have not provided any data for KaiC1 phosphorylation oscillation either in vitro or in vivo. Are those data published anywhere that I do not know?

Answer: In this manuscript, we have focused on the discovery of a new KaiA3 homolog and its connection to KaiB3 and KaiC3. The reviewer is correct that there are no data showing that KaiA1B1C1 is a bona fide oscillator. Because of its high homology with *Synechococcus* sp. PCC 7942 and previous demonstration of KaiA1-dependent phosphorylation of KaiC1, it is likely that the KaiA1B1C1 system behaves similarly to the *Synechococcus* clock. Further evidence came from a study by the Johnson group⁵), which showed that both Kai systems control circadian gene expression. However, this was not a biochemical proof, and it also seems possible that the KaiA1B1C1 system is not capable of sustained oscillation without being coupled to the KaiA3B3C3 system. Unfortunately, the heterologously expressed KaiC1 protein was not stable or homogenous enough to prove that KaiC1 phosphorylation oscillates for at least 24 h. Therefore, we teamed up with Michael Rust and Gopal Pattanayak from the University of Chicago, who were able to provide extended in vivo data on KaiC1 and KaiC3 phosphorylation using high-resolution SDS-PAGE, followed by Western Blot analysis. Using our KaiC1- and KaiC3-specific antibodies, they showed that KaiC1 and KaiC3 abundance and phosphorylation oscillated with a period of approximately 24 h in vivo over the tested time period of three days of constant light conditions, supporting our conclusion that both systems can function as oscillators. Both oscillators appeared to be phase-locked, suggesting that they work together in the cell. The data are shown in Fig. 4 in the current manuscript.

Authors cited the recent work from Johnson lab (Zhao et al, 2023) where a reporter strain was made to show the bioluminescence oscillation. The rhythm in the oscillation in that reporter strain

was gone in *kaiC1* null as well as in *kaiC3* null cells. The circadian clock study in *Synechocystis* varies a lot from lab to lab. For example, Kucho et al., 2005 showed circadian clock exists in *Synechocystis*, however Beck et al, 2014 claimed the absence of circadian rhythm in *Synechocystis*. Similarly, work from the authors' lab could not generate *kaiB2C2* null but Johnson lab could make *kaiB2C2* null cells. Indeed lot of data are dependent on the background strain. That's why it is advisable to check whether the *KaiA1B1C1* cluster can generate an oscillation in *KaiC1* phosphorylation in the background strain (*KaiC1* western blot) that the authors used for this study and also in vitro. Are those the two oscillators in phase or oscillate with different amplitude and/or period?

Answer: The reviewer has a valuable concern regarding the use of strains from different backgrounds in different laboratories. However, it is also possible that some of the diversity in strain behavior is the result of different growth conditions in laboratories or even in the same laboratory. In the revised version of the manuscript, we have now included more data from different laboratory strains that were analyzed in different laboratories. Overall, from these data, we can exclude the possibility that the *KaiA3B3C3* system is only relevant to one particular laboratory strain. Another reason for the use of different background strains was that not every analysis was possible in all *Synechocystis* variants; for example, backscatter measurements were not possible in the PCC-M strain because the cells aggregated in liquid culture, which prevented the measurements. Our revised manuscript reports the data obtained using the following strains.

1. The Chicago strain was analyzed in the Rust lab. This strain was included in an in vivo experiment, showing free-running oscillations over 66 or 72 h for *KaiC1* and *KaiC3* phosphorylation (Fig. 4).

2. The Uppsala strain (obtained from Uppsala University) was analyzed by the Axmann group. This lab recently developed a method to monitor circadian oscillations via backscatter measurements using this strain, and demonstrated that *kaiA1B1C1* deletion abolishes these rhythms⁶. We further re-constructed *kai* gene deletion mutants in this background and showed that *kaiA3B3C3* deletion leads to quick dampening of the rhythms. Upon close inspection, the derived information is in agreement with a recently published paper by the Johnson lab⁵. As shown in Fig. 2 by Zhao et al.⁵, reduced dampening peaks were observed in their Δ *kaiB3C3* mutant strain after release to constant light.

3. The *Synechocystis* PCC-M strain is a motile, glucose-tolerant strain for which the Wilde group generated and analyzed clock-related mutants in previous publications^{2, 3, 4}. We included spot assays for *kai* gene mutants in this strain background. We considered using the same strain to be important for comparison with previously published phenotypes of Δ *kaiA1B1C1* and Δ *kaiC3* using the same method and growth conditions. For this PCC-M strain, we have already demonstrated that *KaiC3* phosphorylates in a *KaiA3*-dependent manner (Fig. 3 in the previous manuscript version). We have moved the data to Fig. 7 to more clearly discuss the observed phenotypes in the *kaiA3* mutants (shown in Fig. 6 and Fig. S11) might go back to the dysregulation of *KaiC3* phosphorylation.

We agree with the reviewer that it is important to check whether oscillations in *KaiC1* phosphorylation also occur in this strain. We performed a Phostag analysis of *KaiC1* phosphorylation, similar to that of *KaiC3*, as shown in Fig. 7a. However, in high-resolution SDS-PAGE, *KaiC1* phosphorylation (Fig. 4a) was easier to interpret than in the Phostag analysis because of fewer cross-reactions with the antibodies. Nevertheless, the Phostag gel analysis showed a similar *KaiC1* phosphorylation pattern compared to *KaiC1* phosphorylation in the wild type used by the Rust laboratory. Interestingly, in both analyses, *KaiC1* phosphorylation had a lower amplitude than *KaiC3* phosphorylation. We did not include the data in the manuscript because the data shown in Fig. 4 cover a longer time period, and we present the data in the PCC-M strain only in the context of spot assays to discuss the effect of *KaiA3* upregulation or downregulation.

Figure 3A and 3C are confusing and can be improved a lot. Please clarify.

Answer: This figure has been rearranged to improve clarity (see the next point).

In fig 3A, authors used purified recombinant proteins (KaiA3, B3 and C3) and checked the circadian oscillation of phosphorylated KaiC3 (PPKaiC3) for 48 hours and claimed KaiC3 phosphorylation showed sustained oscillation with a 24 hrs rhythm (when KaiA3 is either 1.4 μM or 2.8 μM) that is a characteristic feature of a circadian oscillator.

Looks like in Fig 3A, KaiC3 phosphorylation peaked at 6th hour and dephosphorylated (trough) at 18th hour. At 24th hour, the PPKaiC3 is higher than the 18th hour. This suggests the phosphorylation cycle is complete at 18th hour. However, if you look at the phosphorylation peaks at 6th hour and 30th hour, it does look like a 24 hour cycle. But, in that case it would be clear if the experiment was extended up to 54th hour to see the second peak at 54th hour to confirm a 24 hour cycle.

Is the oscillation in PP-KaiC is dying in the second cycle (please look at 18th to 36th hour compared to 0-18th hour)? Is it not a sustained oscillator? Authors have used three replicates (marked by dots) for each time point, if you take the average of all the dots at each time point and plot, does the trace look like an oscillation (from 18th hour onward)? The reason I am bringing this as the amplitude of PPKaiC3 oscillation in the second cycle is very low and the replicates marked by three dots at each time points have a wide range of variation; check the dots at 24th and 30th hour time points.

Answer: Yes, it was difficult to produce high-amplitude oscillations of KaiC3 phosphorylation in vitro. We now describe the observed pattern in more detail and point out that the first peak of in vitro oscillations was higher than the second peak. However, the new in vivo data provided in Fig. 4 had a higher temporal resolution, and the experiment was extended for 68 h. Fig. 4 clearly shows that the phosphorylation cycle of both KaiCs was complete within 24 h and that these oscillations were sustained for at least three cycles.

We implemented some changes to the plot shown in Fig. 3c (previously Fig. 3a). First, as requested, we plotted the average with standard deviations and connected the points by a line instead of showing the akima spline curve. In addition, we directly labeled the curve with the respective KaiA3 concentrations. The curves representing 1.4 μM KaiA3 and 2.8 μM KaiA3 are now highlighted in bold, because they contain more replicates than the other curves and are discussed in more detail in the text.

We have also revised this description in the text. First, we pointed out that no oscillation occurs with excessive or low KaiA3 concentrations. We then focused on the curves with intermediate KaiA3 concentrations and described the first and second peaks in more detail. When describing the KaiA3B3C3 in vitro oscillator, we now repeatedly use the terms *weak* or *dampened* to attribute the reduced amplitude of the second peak. However, we could reconstitute a third peak in vitro, when analyzing KaiA3B3C3 for 60h (now included in Fig 3d) and showed temperature compensation (see below).

Previous Fig. 3c has been changed accordingly and has been moved to Fig. 7. We have moved the figure because we now demonstrate robust in vivo KaiC3 phosphorylation for a longer time period in Fig. 4.

A true circadian clock oscillator is temperature compensated. KaiC3 ATPase activity is not temperature compensated. Will KaiC3 mediated oscillation be considered as circadian? Any

comments on that?

Answer: Yes, temperature compensation is a defining criterion for circadian oscillations. We already know that backscatter oscillations are temperature compensated in *Synechocystis* and are driven by KaiA1B1C1⁶. In this manuscript, we have now included further data showing that KaiA3B3C3 also contributes to these stable oscillations. Similarly, the Johnson lab showed that oscillatory expression from a strong artificial promoter is temperature compensated and that the oscillation is dependent on both, *kaiA1B1C1* and *kaiC3*⁵. Therefore, we hypothesize that the two systems, KaiA1B1C1 and KaiA3B3C3, are interconnected and drive temperature-compensated oscillations.

The reviewer is right that in Wiegard et al.¹, we showed that both ATPase activity and dephosphorylation of KaiC3 (alone) are temperature-dependent. The Q₁₀ value for KaiC3 ATPase activity was 2.4. If the period of KaiC3 phosphorylation displayed the same temperature-dependence, it would change 2.4 fold between 25°C and 35 °C. We investigated the phosphorylation of KaiC3 in the presence of KaiA3 and KaiB3 at 25, 30, and 35°C (see Fig. 3d). Surprisingly, no drastic changes were observed over 48h. The period at 25°C was extended by approx. 6h in comparison to the periods observed at 30°C and 35°C, which is similar to the temperature compensation observed for *Synechococcus* Kai proteins⁷. We can only speculate that the presence of KaiA3 and/or KaiB3 stabilizes temperature compensation.

Figure 3C: Using KaiC3 specific antibody, authors checked the oscillation of P-KaiC3 in *Synechocystis* cells. They mentioned (line 354, page 11) the two prominent bands which are absent in deltaKaiC3 strain most probably reflect two phosphorylation states of KaiC3. They also marked the very upper band (red arrow) as the fully phosphorylated one and one of the lower bands (blue arrow) is the unphosphorylated form. I am confused with the upper band that is marked with PP-KaiC3. The topmost band (marked for PPKaiC3) should not show up in *kaiA3* null cells? Am I wrong? How come KaiC3 gets phosphorylated in absence of KaiA3? If that band is showing up in *kaiA3* null, then that does not represent the true PP-KaiC3 band. So, how that band was used for quantification to check the oscillation in KaiC3 phosphorylation? Again, authors took the ratio of fully phosphorylated KaiC3 (PP-KaiC3) to non-phosphorylated KaiC3 (N-KaiC3) for quantification. But, why the authors only took the PP-KaiC3 band but did not consider both the phosphorylated bands for quantification?

Answer: Wiegard et al.¹ showed that KaiC3 exhibits autophosphorylation activity in vitro in the absence of KaiA. Therefore, it is not surprising that in the *kaiA3* deletion mutant, there is still some phosphorylated KaiC3 (marked with PP-KaiC3), although to a much lesser extent than in the *kaiA3* overexpression strain, and comparable to the lowest phosphorylation level in the WT. We have included a sentence pointing to KaiC3 autophosphorylation (lines 544-545).

Quantification of KaiC3 phosphorylation from Phostag gels was not as easy due to the appearance of up to four bands which partly overlapped with signals detected in the mutant cell extracts. These four bands were not always detected, but at least three bands were visible in all replicates. Furthermore, in the in vitro phosphorylation experiments (Fig. 3a) and in the cell extracts, separated by high-resolution SDS-PAGE (Fig. 4), only three bands were detected. To determine the bands that should be used for quantification, control experiments were performed. We have previously included them in the supplement but have now moved them to the main text (Fig. 7b,c) to allow direct comparison with the time-course analyses (Fig. 7a). Here, we subjected cell extracts and samples from in vitro experiments to Phostag gel to verify that the upper band observed in the cell extracts fitted the PP-KaiC3 band from in vitro phosphorylation (Fig. 7b). Furthermore, we examined in vitro phosphorylated KaiC3 and incubated it with Lambda phosphatase on a Phostag gel (Fig. 7c), demonstrating that this treatment reduced the PP-KaiC3 (P-KaiC3)-

to NP-KaiC3 ratio. However, the middle band(s) were not always visible; therefore, we did not use them for quantification, although we agree that these bands most likely correspond to the single-phosphorylated forms.

We wanted to highlight that even with this different method and neglecting potential single phosphorylated forms, we observed a phosphorylation pattern comparable to that observed in vivo by SDS-PAGE (now included in Fig. 4a). In the (subjective morning) KaiC3 is hypophosphorylated and phosphorylated during the day. The figure has now been modified as described above, and we present the data in context with the spot assays to 1) avoid confusion about different lab strains and 2) confirm that KaiC3 cycles in the particular strain used for phenotypic analysis via spot assays (PCC-M strain).

In this context, analysis of the KaiA3 overexpressor and $\Delta kaiA3$ strain helped us to draw a careful hypothesis as to why $\Delta kaiA3$, $\Delta kaiC3$, and $\Delta kaiA3B3C3$ displayed different phenotypes under certain conditions. The degree of phosphorylation of KaiC3 might be more critical for the functioning of the whole system than the absence of KaiC3 because this might be partly compensated by KaiA1B1C1. Depending on KaiC3 phosphorylation, it may sequester more or less KaiB1, thereby interfering with the KaiA1B1C1 system. Because we do not know the direct input and output components of the KaiA3B3C3 system, we cannot test this hypothesis at present.

Reviewer #2 (Remarks to the Author):

The manuscript entitled “Two circadian oscillators in one cyanobacterium” by Köbler et al, is a worthy attempt to tackle a really interesting question, namely whether two independent circadian oscillators can exist and operate separately in a single cell; in this case, a bacterium. To the knowledge of this Reviewer, this question has only been addressed persuasively once before, namely in experiments on the classic dinoflagellate system where two separate circadian rhythms could be monitored under free-running constant conditions and the researchers found a significant difference in the free-running periods of the two rhythms as their underlying oscillators ran independently (Nature 362:362-364, 1993; which is a reference that the authors inexplicably do not cite).

Answer: We highly appreciate this suggestion from the reviewer and thank you for pointing us to the very interesting paper by Roenneberg and Morse (1993). Although eukaryotic circadian clock systems are very different from cyanobacterial systems in terms of their molecular details, we definitely should have discussed this publication in the context of our paper, as it has almost the same title as our manuscript. This was embarrassing. We have discussed this study in the revised manuscript. Further, we apologize if we were unclear in our previous manuscript. We do not claim that there are two oscillators in one cell that can operate independently of each other. Our data suggest that the oscillators are coupled into an integrated system. Therefore, we changed the title to “Two interconnected circadian oscillators in a cyanobacterial cell”

Unfortunately, the authors of the current manuscript seem to not appreciate how difficult it is to persuasively make the case for two independent oscillators in a single cell. What they show here is that the KaiA3B3C3 proteins are capable of promoting a weak oscillation of KaiC3 phosphorylation in vitro (“weak” because the reported data show a high variability of data points). They then make two key (and unsupported) assumptions:

1. based on the data from the PCC7942 system, the KaiA1B1C1 system can also oscillate by itself in PCC6803 (what’s the evidence for this in PCC6803?).

Answer: We agree with the reviewer that neither we nor others have shown that the KaiABC homologous system of *Synechocystis* is a bona fide oscillator, similar to the *Synechococcus* clock. One of the main obstacles was the relative instability of the

heterologous KaiC1 protein, which prevented us from performing 24 in vitro phosphorylation experiments with KaiC1, KaiA1, and KaiB1 proteins. Evidence that the KaiA1B1C1 system is responsible for circadian oscillations in *Synechocystis* came from a recent publication by the Johnson laboratory⁵ which showed that inactivation of the *kaiA1B1C1* operon and *kaiB3C3* genes abolished circadian gene expression using a new synthetic luminescence reporter. Our admittedly bold statement was based on this study, and it does not mean that the KaiA1B1C1 system is sufficient to generate strong rhythms. We completely agree that this preliminary evidence requires further validation to maintain this statement in the title of our manuscript. Therefore, we have now provided the following new data in the revised manuscript.

1. Due to the lack of useful reporters for both systems, as suggested by this reviewer (see our comment below for suggestion 2.), we used backscattering of the cell cultures to reveal oscillations during growth under constant light after synchronization of the cultures by dilution. Although the reason for this oscillatory behavior of the backscattering signal from cell cultures is not known, it is evident that inactivation of either the Kai system abolishes or alters this characteristic feature. These new data are shown in Fig. 5 and the Results section (The two oscillators together drive circadian backscatter rhythms). Our data and the study by Zhao et al.⁵ imply that the two circadian systems are connected because neither system can drive self-sustained oscillations alone.

2. We were happy to find that the Michael Rust group showed sustained oscillations of KaiC1 and KaiC3 in *Synechocystis* cells over 66 or 72 h under constant conditions using SDS-PAGE. These data imply that the two oscillators are phase-locked (see Fig. 4 and our explanations in the following comments). Please see also our comments to reviewer 1 and below.

2. that the KaiA1B1C1 system does not interact with the KaiA3B3C3 system in vivo to form a composite single oscillator in vivo. I notice from their 2013 paper that (Fig. 4 of Wiegard et al., 2013) from FLAG-tag pulldowns that the systems do not appear to interact in vivo by, e.g., KaiC monomer exchange. This is a good start, but it doesn't eliminate the possibility of interaction between the two systems in vivo by a different mechanism. Moreover, are they sure that their FLAG-tags have not interfered with functionality of the Kai proteins and/or sterically inhibited monomer exchange? Tags can often inhibit function.

Answer: We did not intend to state that the two oscillators are running completely independently because we did see interactions between proteins in previous publications. Using yeast two-hybrid assays, we showed in Wiegard et al.¹ that KaiC1 and KaiC3 can interact and that specifically KaiB1 interacts with KaiC3 and reduces its ATPase activity. Therefore, we argue that neither oscillator functions independently in a single cell. Indeed, we believe that the FLAG-tagged KaiC versions used in the 2013 study⁸ might have interfered with the function of proteins in the cell. Using the new backscattering data, we can now show that the two systems work together (see below). We have rephrased several parts of the manuscript to clarify that we do not see the two oscillators as two completely independent systems. To this end we also chose "The two oscillators together drive circadian backscatter rhythms" as the subheading of the new result section reporting the backscatter analysis.

In order for the authors to persuasively claim that there are two circadian oscillators in PCC6803, they need to show either (or both) of the following:

1. In vitro test: do experiments like those shown in Fig. 3C but where KaiA1B1C1 and KaiA3B3C3 are both mixed in the test tube at the same time and the phosphorylation cycles of KaiC1 and KaiC3 are independently monitored (perhaps with antibodies specific for KaiC1 vs. KaiC3) for enough cycles to show that the free-running periods of the KaiA1B1C1 and KaiA3B3C3 systems are independent. This experiment needs to be done at different temperatures to confirm temperature compensation of both systems; temperature compensation must be demonstrated for both systems----otherwise, they are not circadian oscillators.

Answer: We agree that this would be a straightforward experiment. Unfortunately, owing to technical difficulties related to the instability of the KaiC1 protein, we were unable to perform in vitro experiments. To support our hypothesis that both oscillators are functional, we performed in vivo experiments. In the revised manuscript, we now show the sustained in vivo phosphorylation cycles of KaiC1 and KaiC3 after entrainment under constant conditions over three subjective night and day cycles (Fig. 4). Here, the quantification of the PP-KaiC3 to NP-KaiC3 ratio was more convincing than that of the Phostag gels. The phosphorylation of both Kai proteins cycles with a period of approximately 24 h in the same phase, but with a different amplitude. This further supports our hypothesis that both oscillators are interconnected. Wiegard et al.¹ showed at least for KaiC3 that the ATPase function was not temperature compensated, but we observed temperature compensation of the weak in vitro oscillation when mixing KaiA3 and KaiB3 with KaiC3 (Fig 3d).

In line with this, backscatter oscillations of *Synechocystis* cells are temperature-compensated⁶, and both KaiA1B1C1 and KaiA3B3C3 are required to maintain backscatter oscillations (⁶and new Fig. 5, see also below). Similarly, Zhao et al.⁵ demonstrated clear temperature compensation of oscillations using a luciferase reporter system. As these oscillations are not evident in *kaiC1* or *kaiC3* mutants, we assume that both Kai systems are required to drive circadian gene expression in *Synechocystis*.

2. In vivo test: using a luminescence reporter system with reporter luciferases driven by the two systems, (1) KaiA1B1C1 (using promoters from one of these genes fused to luxAB) and (2) KaiA3B3C3 (using promoters from one of these genes fused to luxAB), the authors might be able to show that these two rhythms run with different periods (by analogy to the aforementioned paper in Nature 362:362-364, 1993). This experiment could be most persuasively accomplished using two-color luciferases (e.g., Plant Cell Physiol. 45:109-13, 2004) where one color tracks the KaiA1B1C1 system while the other color tracks the KaiA3B3C3 system simultaneously.

Answer: This is a very elegant way to verify that Kai systems control their own gene expression in a circadian manner. However, there is no evidence of autoregulation of the *kaiA1B1C1*, *kaiA3B3*, or *kaiC3* promoters⁹. In contrast to *Synechococcus elongatus*, the respective mRNAs are regulated by a number of antisense RNAs and show only low-amplitude cyclic behavior (even under diurnal conditions)⁹. However, the promoters of these operons/genes are too weak to be used in luciferase reporter systems in highly pigmented organisms. Zhao et al.⁵ had to use a very strong photosynthetic promoter to use luciferase as a reporter for circadian gene regulation. Furthermore, at present, we do not know the direct targets of either system. To achieve this, we would first need to determine the output of the KaiC3 oscillator and identify genes that are directly controlled by RpaA, the output regulator of the KaiC1 system. This is something that we would like to do in the future.

Instead, we have included data on the effect of KaiA3B3C3 (and KaiA1B1C1) on the circadian backscatter oscillation of *Synechocystis* cells. Most likely, this oscillation is downstream of the direct output of the oscillators, allowing us to investigate whether and how the two systems work together. Neither oscillator can generate self-sustained backscatter rhythms. We hope that with the new in vivo data, we can convince the reviewers of our manuscript that both Kai systems oscillate simultaneously in *Synechocystis* and that they are interconnected.

References

1. Wiegard A, et al. *Synechocystis* KaiC3 displays temperature- and KaiB-dependent ATPase activity and is important for growth in darkness. *J Bacteriol* **202**, 1-36 (2020).

2. Dörrich AK, Mitschke J, Siadat O, Wilde A. Deletion of the *Synechocystis* sp. PCC 6803 *kaiAB1C1* gene cluster causes impaired cell growth under light-dark conditions. *Microbiology* **160**, 2538-2550 (2014).
3. Köbler C, Schultz SJ, Kopp D, Voigt K, Wilde A. The role of the *Synechocystis* sp. PCC 6803 homolog of the circadian clock output regulator RpaA in day–night transitions. *Molecular Microbiology* **110**, 847-861 (2018).
4. Scheurer NM, *et al.* Homologs of circadian clock proteins impact the metabolic switch between light and dark growth in the cyanobacterium *Synechocystis* sp. PCC 6803. *Frontiers in Plant Science* **12**, 675227 (2021).
5. Zhao C, Xu Y, Wang B, Johnson CH. *Synechocystis*: A model system for expanding the study of cyanobacterial circadian rhythms. *Front Physiol* **13**, 1085959 (2022).
6. Berwanger LC, *et al.* Self-sustained rhythmic behavior of *Synechocystis* PCC 6803 under continuous light conditions in the absence of light-dark entrainment. *bioRxiv*, 2023.2009.2026.559469 (2023).
7. Nakajima M, *et al.* Reconstitution of circadian oscillation of cyanobacterial KaiC phosphorylation in vitro. *Science* **308**, 414-415 (2005).
8. Wiegand A, *et al.* Biochemical analysis of three putative KaiC clock proteins from *Synechocystis* sp. PCC 6803 suggests their functional divergence. *Microbiology* **159**, 948-958 (2013).
9. Beck C, *et al.* Daily expression pattern of protein-encoding genes and small noncoding RNAs in *Synechocystis* sp. strain PCC 6803. *Applied and Environmental Microbiology* **80**, 5195-5206 (2014).

Reviewer #2 (Remarks to the Author):

The manuscript entitled “Two interconnected circadian oscillators in a cyanobacterium” by Köbler et al, is a worthy attempt to tackle a really interesting question. I still believe that the authors seem to not appreciate how difficult it is to persuasively make the case for two oscillators in a single cell. The authors attempt to address the problem by adding the word “interconnected” to the title. Unfortunately, this is not an adequate “fix” to the problem. The authors appear to be confused about the distinction between OSCILLATOR and RHYTHM. Their observation that there are rhythms of both KaiC1 AND KaiC3 phosphorylation does not mean that there are two oscillators. It is possible that in vivo, C1 and C3 monomers exchange into and out of composite KaiC hexamers. Or that one phosphorylation rhythm is driven by the other. And there are other ways to envision the current data of the relationship between C1 and C3 that do not require two separate oscillators (interconnected or independent).

I was about to reject the manuscript for failing to adequately address the concerns of my previous review, but then I noticed something interesting in the authors’ new backscattering data (Fig. 5) that the authors don’t comment upon. The authors focus upon amplitude and phase (both of which could be affected by output components without a change in properties of the driving oscillator), but they appear to have missed the significance of the change of period in the damped oscillation expressed by the d-A3B3C3 strain. In WT, the period is approximately 24 h, but in the d-A3B3C3 strain, it appears to be approximately 21 h (Fig. 5d). This is a very important result, because it indicates that----while the A1B1C1 oscillator is the predominant driver of the backscattering rhythm----the A3B3C3 “oscillator” plays a subsidiary role in determining period, probably in a coupled oscillator system. I find these data of Fig. 5d to be the most persuasive evidence for the authors’ contention that there is a second oscillator (encoded by A3B3C3) operating in a coupled system with the primary oscillator (encoded by A1B1C1).

To adequately address this point, the authors need to reanalyze and present the data of Figure 5 in the context of PERIOD for all the strains shown. As I said, the data of panel d appear to show a period of ~24 h for WT and of ~21 h for d-A3B3C3, but are the authors really sure that there is not a long-period oscillation in the d-A1B1C1 strain? If so, it could be additional evidence that an oscillator encoded by the A3B3C3 genes can cycle in the absence of KaiA1B1C1 genes and drive very weak, long-period rhythms of backscattering. I look forward to seeing the results of those analyses and the authors’ discussion of their significance.

Incidentally, the authors contend in their 2023 BioArchives preprint that this backscattering rhythm is related to glycogen metabolism, but they do not mention that interpretation in this ms (as they should) and acknowledge publications from other labs that have reported rhythms of glycogen content in cyanobacteria.

Alternatively, I suggest that this rhythm of backscattering might be an indicator of rhythms of cell size/cell number/cell division/growth. If so, the authors should acknowledge and reference prior papers showing cell size/division rhythms in PCC 7942:

Mori et al. PNAS 93: 10183-10188 (1996)

Kondo et al. Science 275: 224-227 (1997)

We are very grateful to Reviewer 2, who raised these important concerns and provided helpful suggestions. You will find our answers in bold following the respective remark of the reviewer.

Answers to the reviewer comments

Reviewer #2 (Remarks to the Author):

The manuscript entitled “Two interconnected circadian oscillators in a cyanobacterium” by Köbler et al, is a worthy attempt to tackle a really interesting question. I still believe that the authors seem to not appreciate how difficult it is to persuasively make the case for two oscillators in a single cell. The authors attempt to address the problem by adding the word “interconnected” to the title. Unfortunately, this is not an adequate “fix” to the problem. The authors appear to be confused about the distinction between OSCILLATOR and RHYTHM. Their observation that there are rhythms of both KaiC1 AND KaiC3 phosphorylation does not mean that there are two oscillators. It is possible that in vivo, C1 and C3 monomers exchange into and out of composite KaiC hexamers. Or that one phosphorylation rhythm is driven by the other. And there are other ways to envision the current data of the relationship between C1 and C3 that do not require two separate oscillators (interconnected or independent).

Answer: We agree with the reviewer that it is not clear from the data whether in vivo both the KaiA1B1C1 and KaiA3B3C3 proteins independently meet the oscillator criterion in the meaning of a system that can autonomously create self-sustained rhythms. Therefore, we changed the title of the manuscript to “Two KaiABC systems control circadian oscillations in one cyanobacterium”, and discuss our data accordingly. However, as outlined below, the suggestion to re-analyze the data was very helpful to see that both systems are still able to drive rhythms alone. After the deletion of the *kaiA3B3C3* cluster, the oscillations were dampened. To avoid confusion, we added a general statement in the discussion in lines 666-667: “In this work, we broadly define an oscillator to include systems that may be dampened but nevertheless have a natural frequency.”

We have also revised the manuscript to indicate more clearly when we were writing about phosphorylation rhythms. E.g.:

Line 379: “KaiC1 and KaiC3 phosphorylation are phase-locked in *Synechocystis* cells”

Line 389-390: “KaiC1 phosphorylation displayed stable ~24 h rhythms which were phase-locked with the KaiC3 phosphorylation rhythm.”

I was about to reject the manuscript for failing to adequately address the concerns of my previous review, but then I noticed something interesting in the authors’ new backscattering data (Fig. 5) that the authors don’t comment upon. The authors focus upon amplitude and phase (both of which could be affected by output components without a change in properties of the driving oscillator), but they appear to have missed the significance of the change of period in the damped oscillation expressed by the d-A3B3C3 strain. In WT, the period is approximately 24 h, but in the d-A3B3C3 strain, it appears to be approximately 21 h (Fig. 5d). This is a very important result, because it indicates that---while the A1B1C1 oscillator is the predominant driver of the backscattering rhythm----the A3B3C3 “oscillator” plays a subsidiary role in determining period, probably in a coupled oscillator system. I find these data of Fig. 5d to be the most persuasive evidence for the authors’ contention that there

is a second oscillator (encoded by A3B3C3) operating in a coupled system with the primary oscillator (encoded by A1B1C1).

To adequately address this point, the authors need to reanalyze and present the data of Figure 5 in the context of PERIOD for all the strains shown. As I said, the data of panel d appear to show a period of ~24 h for WT and of ~21 h for d-A3B3C3, but are the authors really sure that there is not a long-period oscillation in the d-A1B1C1 strain? If so, it could be additional evidence that an oscillator encoded by the A3B3C3 genes can cycle in the absence of KaiA1B1C1 genes and drive very weak, long-period rhythms of backscattering. I look forward to seeing the results of those analyses and the authors' discussion of their significance.

Answer: Thank you for your very helpful suggestion. Based on this idea, we analyzed the backscatter rhythms in more detail. To compare the periods of the backscatter rhythms between the strains, we first attempted to fit a cosine function for a simple harmonic oscillation to the normalized backscatter signals. The wild-type signal fitted well to the function describing a harmonic oscillation, and we could determine a period of approximately 26h, which is in agreement with the period determined from peak analysis performed by Berwanger *et al.*¹ The amplitude of backscatter changes in the $\Delta kaiA1B1C1$ strain was extremely low, but could nevertheless be described by the same equation for a simple harmonic oscillation in all three experiments. As pointed out by the reviewer, this period was extended to approximately 33.3 hours (see Figure S10b for a list of parameters and periods).

The backscatter rhythms in the *kaiA3B3C3* deletion strain cannot be accurately described by simple harmonic oscillations without dampening. In general, all mutants in which *kaiC3* was deleted displayed slightly different dampening patterns in the replicate experiments (Fig 5f and Fig. S10), although wild-type data were highly reproducible. Therefore, we believe that the variations after 24h indicate a loss of robustness in all mutants in which *kaiC3* was deleted. Because the simple harmonic cosine function was not sufficient to determine the period from the strains that displayed dampening, we identified the first trough and peak and multiplied the distance by two to approximate the period of the first cycle. This method was chosen to measure the period length because it was sometimes impossible to determine the later peaks using the same parameters for each strain. When we performed this analysis for the wild type, the period of the first backscattering cycle was comparable to the period determined from the cosine fit ($26.9 \pm 0.75h$ and $26.46h \pm 0.34h$, respectively). In the *kaiA3B3C3* deletion strain and in all other strains in which *kaiC3* was deleted, the period was reduced to approx. 21h (Fig. 5j).

We have included the new analysis in Figures 5 and S10, and described the results starting from line 431.

We added the following hypothesis in lines 444-448: "Altogether, the backscattering data imply that the KaiA1B1C1 oscillator mainly drives circadian rhythms, but requires KaiA3B3C3 to maintain the period and amplitude. On the other hand, KaiA3B3C3 may be able to drive low-amplitude oscillations but requires coupling to KaiA1B1C1 to maintain a circadian period and ensure a high amplitude.

The significance is discussed in the context of the published data in the section starting with line 781.

In addition, we have added the following findings to the abstract: "Deletion of either system altered the period of the backscattering rhythm."

We also want to highlight that the cosine fits demonstrated again that the deletion of *kaiA1B1C1* and *kaiB3* had comparable effects on the backscatter rhythms. In contrast to the strains with *kaiC3* or *kaiA3* deletions, Δ *kaiB3* mutants displayed low-amplitude oscillations, which could be described by the equation for a simple harmonic oscillation without dampening. The period was further extended compared with that observed for the *kaiA1B1C1* mutant strain. We hypothesize that coupling of the oscillators might occur via KaiB proteins, as outlined in the model in Figure 8.

Incidentally, the authors contend in their 2023 BioArchives preprint that this backscattering rhythm is related to glycogen metabolism, but they do not mention that interpretation in this ms (as they should) and acknowledge publications from other labs that have reported rhythms of glycogen content in cyanobacteria.

Answer: We added the hypothesis and literature to the discussion in lines: 776-780

Alternatively, I suggest that this rhythm of backscattering might be an indicator of rhythms of cell size/cell number/cell division/growth. If so, the authors should acknowledge and reference prior papers showing cell size/division rhythms in PCC 7942:

Mori et al. PNAS 93: 10183-10188 (1996)

Kondo et al. Science 275: 224-227 (1997)

Answer: We agree that circadian regulation of cell division is an intuitive hypothesis to explain backscatter oscillations. To date, we do not have any data to support this hypothesis. Nevertheless, we added the possibility and literature to the discussion (lines 780-781)

Reviewer #2 (Remarks to the Author):

The revised manuscript by Koebler and collaborators includes many of my suggestions from the original review.

From the new data and analyses in Figure 5, it appears that kaiC3 is critical for maintaining a wild-type period but kaiA3 and kaiB3 are not important as far as period is concerned. On the other hand, kaiA3 and kaiB3 are important for amplitude and sustained rhythmicity. These data do not persuasively support the authors' original conclusion & title of "Two circadian oscillators in one cyanobacterium," but the revised title and conclusion "Two KaiABC systems controlling circadian oscillations in one cyanobacterium," is defensible from the authors' data.

This conclusion and the identification of the NarL-like kaiA3 are the major new insights of this study.

Minor suggestion for the final text:

The statement in the Introduction by the authors, "In contrast to the study by Zhao et al.17, deletion of kaiC3 in the motile *Synechocystis* strain (PCC-M29) used in our study had no effect on growth under light/dark (LD) cycles" is an exaggeration since the effect in the Zhao study is small (barely significant). I suggest that the authors rephrase this sentence to read " In contrast to the study by Zhao et al.17 that reported a small growth defect in *Synechocystis* in which kaiC3 has been deleted, our studies with the motile *Synechocystis* strain (PCC-M29) showed no such effect on growth under light/dark (LD) cycles."

Reviewer 2:

Minor suggestion for the final text:

The statement in the Introduction by the authors, "In contrast to the study by Zhao et al.17, deletion of kaiC3 in the motile *Synechocystis* strain (PCC-M29) used in our study had no effect on growth under light/dark (LD) cycles" is an exaggeration since the effect in the Zhao study is small (barely significant). I suggest that the authors rephrase this sentence to read " In contrast to the study by Zhao et al.17 that reported a small growth defect in *Synechocystis* in which kaiC3 has been deleted, our studies with the motile *Synechocystis* strain (PCC-M29) showed no such effect on growth under light/dark (LD) cycles."

Answer:

We are very grateful to Reviewer 2, who carefully reviewed the revised version of our paper. In the revised version of our paper, we have made the suggested change (lines 92-94) in the final text.